# Non-canonical dihydrolipoyl transacetylase promotes chemotherapy resistance via mitochondrial tetrahydrofolate signaling

Jung Seok Hwang[1], JiHoon Kang[1], Jaehyun Kim[1], Kiyoung Eun[1], Sophia West[2], Hannah E. Bacho[1], Vanessa Avalos[1], Sydney Shuff[1], Dong M. Shin[1], Nabil F. Saba [1], Kelly R. Magliocca[3], Cheng-Kui Qu [4], Haian Fu [1,2], Suresh S. Ramalingam [1], Andrey A. Ivanov [2], Taro Hitosugi [5] & Sumin Kang [1] ✉

Chemotherapy is often a primary treatment for cancer. However, resistance leads to therapeutic failure. Acetylation dynamics play important regulatory roles in cancer cells, but the mechanisms by which acetylation mediates therapy resistance remain poorly understood. Here, using acetylome-focused RNA interference (RNAi) screening, we find that acetylation induced by mitochondrial dihydrolipoyl transacetylase (DLAT), independent of the pyruvate dehydrogenase complex, is pivotal in promoting resistance to chemotherapeutics, such as cisplatin. Mechanistically, DLAT acetylates methylenetetrahydrofolate dehydrogenase 2 (MTHFD2) at lysine 44 and promotes 10-formyl-tetrahydrofolate (10-formyl-THF) and consequent mitochondrially encoded cytochrome c oxidase II (MT-CO2) induction. DLAT signaling is elevated in cancer patients refractory to chemotherapy or chemoimmunotherapy. A decoy peptide DMp39, designed to target DLAT signaling, effectively sensitizes cancer cells to cisplatin in patient-derived xenograft models. Collectively, our study reveals the crucial role of DLAT in shaping chemotherapy resistance, which involves an interplay between acetylation signaling and metabolic reprogramming, and offers a unique decoy peptide technology to overcome chemotherapy resistance.

Chemotherapy continues to serve as a predominant and highly effective front-line therapeutic option, especially for many cancer patients who are not eligible for or exhibit a limited response to immunotherapy[1,2]. Cisplatin, a metal platinum-based drug that cross-links to DNA and impairs DNA synthesis, is one of the most compelling chemotherapy agents widely used in treating various cancers[3–5]. Cisplatin and other chemotherapy agents such as carboplatin, pemetrexed, gemcitabine, etoposide, and paclitaxel are therapeutic options

for patients with lung, head and neck, cervical, and ovarian cancers[6–8]. The combination of these chemotherapy agents with immunotherapy has shown promising therapeutic outcomes and has become a new option in the treatment of various types of cancer[9]. However, the development of resistance in cancer patients leads to relapse, ultimately resulting in therapeutic failure[10,11]. Numerous studies have explored resistance models, including reduced drug uptake, target pathway alteration, compensatory pathway activation for cell survival,

[1]Department of Hematology and Medical Oncology, Winship Cancer Institute of Emory, Emory University School of Medicine, Atlanta, GA, USA. [2]Department of Pharmacology and Chemical Biology, Emory University School of Medicine, Atlanta, GA, USA. [3]Department of Pathology & Laboratory Medicine, Emory University School of Medicine, Atlanta, GA, USA. [4]Department of Pediatrics, Emory University School of Medicine, Atlanta, GA, USA. [5]Department of Oncology, Division of Oncology Research, Mayo Clinic, Rochester, MN, USA. ✉e-mail: smkang@emory.edu

and the regulation of the tumor microenvironment[12]. These factors may jointly act to induce therapeutic resistance in cancers, but effective and non-toxic regimens to overcome resistance remain elusive[13–16]. Therefore, the identification of an effective key resistance driver and potent targeted agents is urgently needed.

Acetylation modification is a post-translational modification involved in many biological processes in cells. Abnormal acetylation levels may lead to the occurrence or progression of cancer[17]. Histone acetylation is known as a determinant of chemoresistance[18,19]. However, few studies have comprehensively linked acetylation modification to chemotherapy resistance, specifically how acetylation induces metabolic rearrangement and confers resistance. Understanding and targeting a critical acetylation switch that drives chemotherapy resistance may decisively improve the therapeutic and curative outcomes of chemotherapy. Through an acetylome-wide RNAi screen and proteomics analyses, here we identified dihydrolipoamide acetyltransferase (DLAT) and the unique acetylation of methylenetetrahydrofolate dehydrogenase 2 (MTHFD2) induced by DLAT as an essential driver of cisplatin resistance in lung, head and neck, and cervical cancers. DLAT is the component of the pyruvate dehydrogenase complex (PDC) that catalyzes the conversion of pyruvate to acetyl-CoA in the mitochondria, which is a key reaction that occurs at the junction of the citric acid cycle and glycolysis[20–22]. Human PDC is composed of three major enzymes: pyruvate dehydrogenase (PDH; E1), dihydrolipoyl transacetylase (DLAT; E2), and dihydrolipoyl dehydrogenase (DLD; E3)[23–25]. Since PDH catalyzes the rate-limiting step, converting pyruvate to acetyl-CoA, PDH activity determines the PDC flux rate and is tightly regulated by reversible phosphorylation[26,27]. Correlation studies revealed that DLAT is upregulated in human cancers, including hepatocarcinoma and gastric and pancreatic cancers[28–30]. For instance, high expression of DLAT positively correlated with drug resistance, immune infiltration, and pancreatic cancer progression, revealing the value of DLAT as a therapeutic target[29].

Methylenetetrahydrofolate dehydrogenase 1 and 2 (MTHFD1/2) are enzymes essential in the folate-mediated one-carbon metabolism pathway. While cytosolic MTHFD1 is ubiquitously expressed in tissues, MTHFD2 is highly abundant in the mitochondria of transformed cells[31]. A meta analysis including 19 cancer types emphasized MTHFD2 as one of the most consistently upregulated metabolic genes in human cancers, and it is associated with poor prognosis in cancers, including breast, liver, and pancreatic cancers[32–35]. MTHFD2 catalyzes the conversion from 5,10-methylene-tetrahydrofolate (THF) to 5,10-methenyl-THF and then to 10-formyl-THF, which subsequently supplies formate. Although the significance of MTHFD2 in human cancer is well studied, the precise mechanism by which MTHFD2 is activated to confer chemotherapy resistance is unclear. MTHFD2 has been tied to deacetylases SIRT3 in colorectal cancer and SIRT4 in immune evasion in pancreatic cancer[36–38]. However, the acetylation status of MTHFD2 and the underlying mechanism in other cancers remain unknown.

Inhibitors targeting components of PDC include a non-selective PDK inhibitor, dichloroacetate (DCA)[39]. However, to the best of our knowledge, no specific inhibitor is available to target non-classical DLAT signaling that is independent of PDC. LY345899 and DS18561882 have been reported to be MTHFD2 inhibitors, and are a folate analog and a tricyclic coumarin-based compound that competitively binds to the substrate binding site[40–42]. However, these inhibitors are unselective, concurrently inhibiting the ubiquitous isozyme MTHFD1 and thus posing a potential safety risk; therefore, further efforts are still needed to develop potent and specific MTHFD2 inhibitors[43,44].

In this work, we report the non-classical role of DLAT in driving chemotherapy resistance. We uncover the molecular mechanism by which DLAT activates its unique substrate MTHFD2 and provides resistance through 10-formyl-THF in human cancers. We also offer a peptide-based DLAT-MTHFD2 inhibitor as a promising therapeutic option to improve chemotherapy outcomes.

## Results

### Acetyltransferase DLAT confers cisplatin resistance in human cancers

To screen for acetylation-associated factors that are pivotal for cisplatin-resistant cancer survival, we performed an RNAi-based cisplatin sensitivity assay using a customized shRNA library that contains effective suppressors of 74 acetylation-related genes, including 50 acetyltransferases and 24 deacetylases. The RNAi library contains 328 shRNA clones, and each gene was targeted by 2 ~ 8 clones (Fig. 1a). Eleven genes, including DLAT, were selected as unexplored lead hits with cisplatin-sensitizing potential and were further evaluated in A549$^{cisR}$, H1299, and KB-3-1$^{cisR}$, and two acetyltransferases, dihydrolipoamide S-acetyltransferase (DLAT) and lysine acetyltransferase 2 A (KAT2A), were selected as common resistance drivers (Fig. 1b). DLAT was selected as the target when the clinical significance was evaluated by overall survival comparison between high expression and low expression groups in 9 cancers in which cisplatin is often applied (Fig. 1c). The screening results for DLAT were confirmed as DLAT knockdown sensitized various cancer cell types to cisplatin treatment (Fig. 1d). DLAT loss and sublethal doses of cisplatin attenuated colony-forming ability (Fig. 1e) and cell viability and induced apoptotic cell death in KB-3-1 and A549 cells that are resistant to cisplatin. Similar effects on cisplatin sensitivity were also observed in PCI-37B and H1299 cells (Fig. 1f). Cisplatin-resistant or cisplatin-less sensitive (cis$^R$) cells were generated from parental cells by continuous exposure to cisplatin (cisplatin IC$_{50}$ values: KB-3-1: 8.4 μM, KB-3-1$^{cisR}$: 60.5 μM, A549: 2.3 μM, A549$^{cisR}$: 17.6 μM), and optimal sublethal doses of cisplatin for each cell line were used throughout the study[45]. Next, we functionally confirmed the DLAT effect in vivo. Xenografted tumors with DLAT knockdown and sublethal dose of cisplatin treatment showed dramatically decreased tumor volume, mass, and proliferation (Fig. 1g-j). DLAT is known to be induced by a copper-binding ionophore elesclomol and sensitizes docetaxel sensitivity in prostate cancer[46]. However, we found that various doses of elesclomol did not alter the DLAT levels in the cancer cells we tested (Supplementary Fig. 1). These data indicate that DLAT mediates cisplatin resistance, its inhibition can enhance cancer cell sensitivity to cisplatin, and the contribution can be cancer-type dependent.

### Cisplatin-resistant cancer survival requires DLAT activity that is independent of PDC

The genetic inhibition study performed only suggests that the DLAT protein is crucial for cisplatin resistance. We tested whether the acetyltransferase activity of DLAT is required for the resistance. An enzyme-dead mutant form of DLAT was established by deleting the catalytic domain B (ΔB; 417-647 amino acids) in DLAT. Rescue expression of wildtype, but not ΔB DLAT, restored the cisplatin resistance decreased by DLAT loss, suggesting that the acetyltransferase activity is needed for DLAT to offer cisplatin-resistant survival potential to cancer cells (Fig. 2a). DLAT is a component of the multi-enzyme PDC. Therefore, we examined whether DLAT confers cisplatin resistance by modulating PDC activity. PDC activity, which was assessed by PDH1 phosphorylation and reaction, was abolished when DLAT, the subunit E2 of PDC, was lost. However, distinct from the decrease observed in cell viability, DLAT loss and cisplatin treatment did not further alter either PDC activity or the interaction between PDC components (Fig. 2b–d). Pharmacological inhibition of PDC using dichloroacetate (DCA) had no significant effect on the survival of cisplatin-resistant cancer cells (Fig. 2e). In addition, overexpression of ACSS1, which converts acetate into the PDC/DLAT product acetyl-CoA in the mitochondria, did not rescue the decreased cisplatin resistance caused by DLAT loss (Fig. 2f). Furthermore, the knockdown of other PDC subunits, E1 or E3, did not sensitize the cells to cisplatin (Fig. 2g, h). These results indicate that the contribution of DLAT to cisplatin resistance is independent of PDC.

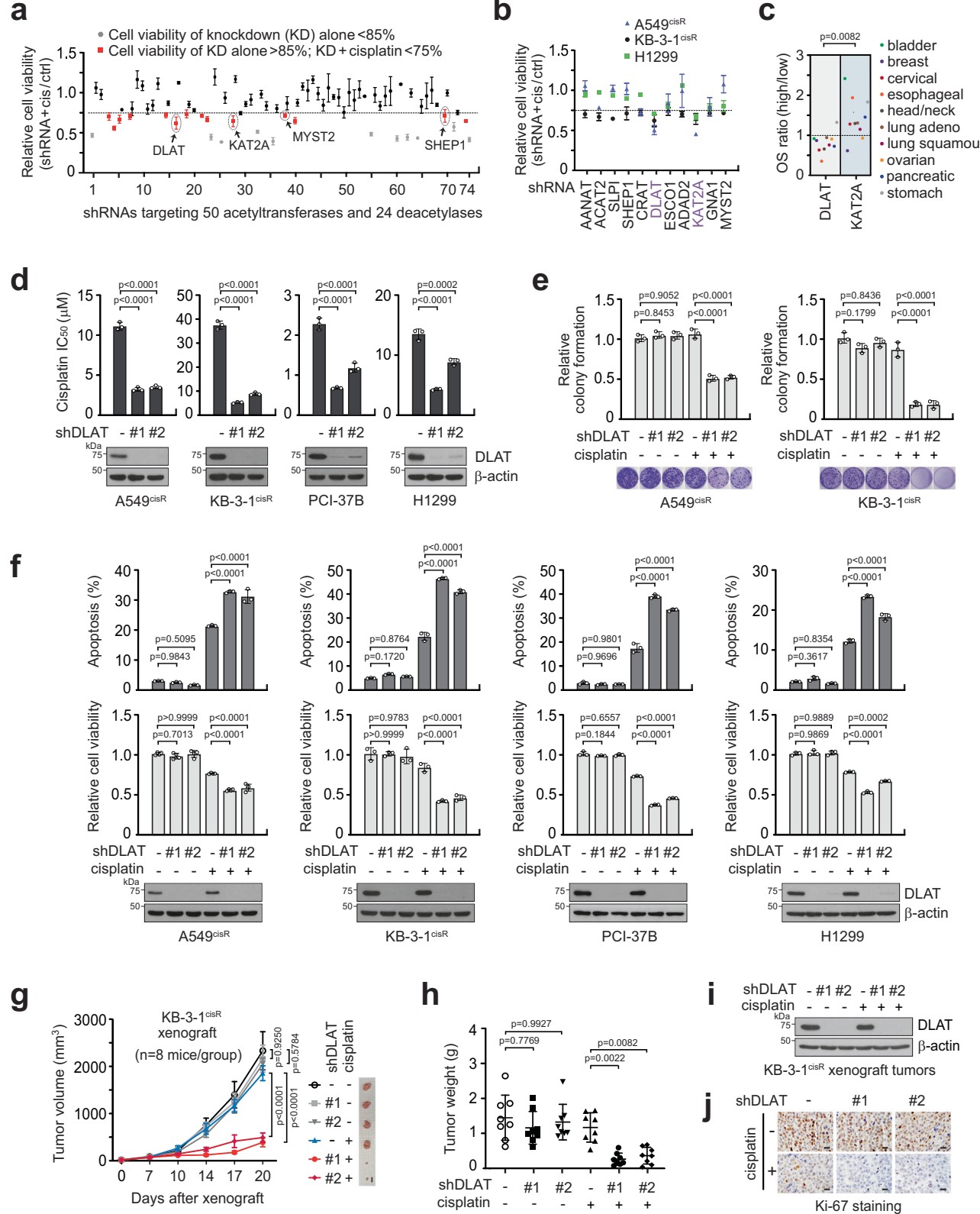

## DLAT controls mitochondrial ROS produced by chemotherapy agents

Cisplatin's mode of action involves DNA lesion generation followed by activation of the DNA damage response and induction of apoptosis, and resistance can occur at different levels. The loss of DLAT resulted in cisplatin-induced apoptosis, but target downregulation of DLAT did not impact cisplatin-DNA adduct accumulation or cisplatin-induced

DNA damage, suggesting that DLAT is involved in a post-target but not a pre- or on-target resistance mechanism (Fig. 3a). To investigate whether DLAT offers metabolic gain to support cisplatin resistance, we examined alterations in bioenergetics, protein and RNA synthesis, and redox levels in cells with DLAT knockdown and cisplatin. Impairment of DLAT did not alter the levels of energy, protein, or RNA, but elevated ROS levels and lowered NADPH level and reduced/oxidized

**Fig. 1 | Customized acetylation-focused RNAi screen identifies DLAT as a critical driver of cisplatin resistance in cancer. a** An initial screening examining the effect of loss of 50 acetyltransferases and 24 deacetylases on cisplatin response. Each gene was targeted by pooled shRNA virus infection. KB-3-1$^{cisR}$ cells transduced with lentiviral shRNA were treated with a sublethal dose of cisplatin (5 μg/ml). Candidates exhibiting low cell viability upon knockdown (<85%; gray) were excluded. **b** Secondary screening utilized the top 11 genes identified from the primary screen in A549$^{cisR}$, KB-3-1$^{cisR}$, and H1299. Two candidates that induced over 25% cell death upon gene knockdown and cisplatin treatment are highlighted in purple. **c** Overall survival (OS) ratios for high versus low expression of the top 2 genes, DLAT and KAT2A, in 10 cancer types were determined using TCGA-based Kaplan-Meier Plotter. Pan-cancer RNA-sequencing data from 3565 patients was stratified by medians. **d** Effect of DLAT knockdown on cisplatin response was determined by cisplatin IC$_{50}$. Cells were treated with sublethal doses of cisplatin (A549$^{cisR}$ 2 μg/ml, KB-3-1$^{cisR}$ 5 μg/ml, PCI-37B 1 μg/ml, H1299 5 μg/ml) for 48 h. **e** Colony formation potential of cancer cells with cisplatin treatment and DLAT knockdown. **f** Cisplatin-induced apoptotic cell death (upper) and cell viability (lower) in cancer cells without or with DLAT knockdown. **g–j**, DLAT knockdown effect on cisplatin-resistant tumor growth. Mice xenografted with KB-3-1$^{cisR}$ were treated with vehicle control or cisplatin (5 mg/kg) twice a week. Tumor size (**g**), tumor weight (**h**), and the expression level of DLAT in tumors (**i**) are shown. Tumor proliferation is assessed by Ki-67 IHC, and representative images of the staining are shown for each group (**j**). Scale bars shown in (**g**) and (**j**) represent 10 mm and 50 μm, respectively. Data are mean ± SD from 3 technical replicates for (a) and 3 independent biological replicates for (**b**, **d–f**). $n = 8$ per group for (**g**) and (**h**), and the error bars indicate SEM for (**g**) and SD for (**h**). $P$ values were determined by paired two-tailed Student's $t$-test for (**c**), two-way ANOVA (**g**), and one-way ANOVA for the rest. Source data are provided as a Source Data file.

glutathione (GSH/GSSG) ratio in cisplatin-treated cells (Fig. 3b, c and Supplementary Fig. 2). Treatment with an antioxidant N-acetyl-cysteine (NAC) significantly lowered cisplatin-mediated ROS and apoptosis, thereby recovering cell viability in DLAT knockdown cells (Fig. 3d). Mitochondrial-specific antioxidant mito-TEMPO, but not ectopic expression of the cytosolic ROS scavenger, catalase, reversed the elevated apoptosis and attenuated cell viability induced by DLAT loss and cisplatin (Fig. 3e, f). These data suggest that DLAT promotes cisplatin-resistant cancer cell survival by managing mitochondrial ROS rather than cytosolic ROS. While ROS influences senescence, autophagy, and cell cycle, neither loss of DLAT nor NAC treatment affected these phenotypes, suggesting that DLAT-controlled ROS primarily supports cisplatin resistance through anti-apoptotic mechanisms (Supplementary Fig. 3). Assessment of a panel of apoptotic factors showed that the knockdown of DLAT in cisplatin-treated cells specifically led to the attenuation of anti-apoptotic factor Bcl-xL (Fig. 3g). In contrast, control of elevated ROS with NAC in these cells partially restored Bcl-xL expression (Fig. 3h). These data suggest that DLAT reduces ROS levels, thereby inducing Bcl-xL expression and contributing to the suppression of apoptotic cell death. Moreover, impairment of DLAT led to ROS accumulation and sensitized cancer cells to various chemotherapy agents, including carboplatin, gemcitabine, etoposide, pemetrexed, and paclitaxel, but not to a molecularly targeted inhibitor, erlotinib, suggesting that DLAT specifically controls chemotherapy-induced ROS and cell death (Fig. 3i).

## DLAT binds to, acetylates, and activates MTHFD2 to provide cisplatin resistance

To investigate how DLAT communicates with cellular proteins to manage mitochondrial ROS and confer cisplatin resistance, we performed interaction proteomics profiling using A549$^{cisR}$ cells with or without DLAT. Mass spectrometry analysis showed that DLAT binds to mitochondrial proteins, including methylenetetrahydrofolate dehydrogenase 2 (MTHFD2), mitochondrial ribosomal protein L1 (MRPL1), peroxiredoxin 3 (PRDX3), and RNA polymerase mitochondrial (POLRMT) and the interaction between DLAT and these potential interactors was confirmed by co-immunoprecipitation (Fig. 4a–c and Supplementary Fig. 4a). Target inhibition studies of these interactors revealed that loss of MTHFD2 or POLRMT, but not PRDX3 or MRPL1 enhanced cisplatin-induced cell death, mimicking the DLAT effect (Fig. 4d and Supplementary Fig. 4b–d). MTHFD2 is a mitochondrial redox-regulating enzyme and could be a potential downstream effector contributing to DLAT-associated redox balance in cells. Pharmacological inhibition of MTHFD2 with DS18561882 and LY345899 sensitized cancer cells to cisplatin treatment, suggesting that the activity of MTHFD2 is required for cells to confer resistance to cisplatin (Fig. 4e, f). Moreover, the interaction between DLAT and MTHFD2 is independent of PDC and DLAT and MTHFD2 co-localize in the mitochondria, as confirmed in cisplatin-treated cancer cells (Fig. 4g–k).

To investigate the functional link between DLAT and MTHFD2, we performed an in vitro DLAT acetylation assay using a recombinant MTHFD2 as a substrate. DLAT directly acetylated MTHFD2 (Supplementary Fig. 5a). Acetyl-proteomics analysis demonstrated that K44, K50, and K286 in MTHFD2 were acetylated by DLAT. Among them, the K44 and K50 sites were reported to be acetylated in our acetyl-proteomics data and the publicly available PTM database (Fig. 5a and Supplementary Fig. 5b). A coupled in vitro DLAT acetylation and MTHFD2 activity assay revealed that DLAT-induced acetylation of MTHFD2 activates MTHFD2 wildtype and acetylation-deficient mutant K50R, while the activity was unaltered in MTHFD2 K44R, where the acetylation at K44 is restrained (Fig. 5b). The specificity of the customized antibody generated against acetyl-K44 MTHFD2 was confirmed using MTHFD2 and DLAT variants in cells (Fig. 5c, d). To further understand how DLAT acetylates MTHFD2, we mutated S475 and M570 in DLAT, the residues that may be involved in MTHFD2 acetylation. The mutational analysis revealed that the S475 mutation in DLAT, but not the M570 mutation, significantly reduced K44 MTHFD2 acetylation and activation, indicating that serine 475 of DLAT is crucial for the acetylation of MTHFD2 (Supplementary Fig. 5c). The acetylation of MTHFD2 at K44 by DLAT was independent of other components of the PDC (Fig. 5e and Supplementary Fig. 5d). Moreover, acetyl-mimetic MTHFD2 mutant K44Q, but not acetyl-deficient MTHFD2 mutant K44R, significantly rescued the decreased cisplatin-resistant cell survival and the elevated cellular ROS levels seen in DLAT knockdown cells (Fig. 5f, g left). Furthermore, the acetyl-mimetic MTHFD2 mutant K44Q cells showed a significant recovery in antioxidant indicators, such as the NADPH level and GSH/GSSG ratio (Fig. 5g middle and right). Although MTHFD2 is known to be deacetylated by SIRT3 in colon cancer, the loss of DLAT did not affect cisplatin sensitivity or acetylate MTHFD2 at K44 in colon cancer cell lines, suggesting that DLAT modulates neither acetylation regulation nor cisplatin sensitivity in colon cancer and that the role of DLAT in chemotherapy resistance may be different depending on cancer types (Supplementary Fig. 6). These data suggest that acetylation of MTHFD2 by DLAT at K44 activates MTHFD2 to manage ROS induced by chemotherapy agents, such as cisplatin.

## DLAT-MTHFD2 contributes to cisplatin resistance through 10-formyl-THF

MTHFD2 catalyzes the conversion of 5,10-methylene-THF to 10-formyl-THF, eventually generating formate in the mitochondria. To investigate whether these metabolites support cisplatin resistance, cells lacking MTHFD2 acetylation were replenished with the metabolic products of MTHFD2 in the folate cycle. Restoring cells with physiological concentrations of 10-formyl-THF, but not formate, reversed the attenuated cell viability and elevated apoptosis that was induced by cisplatin and DLAT loss (Fig. 6a, b). Moreover, decreased cell viability and elevated apoptosis due to acetyl-deficient MTHFD2 K44R were

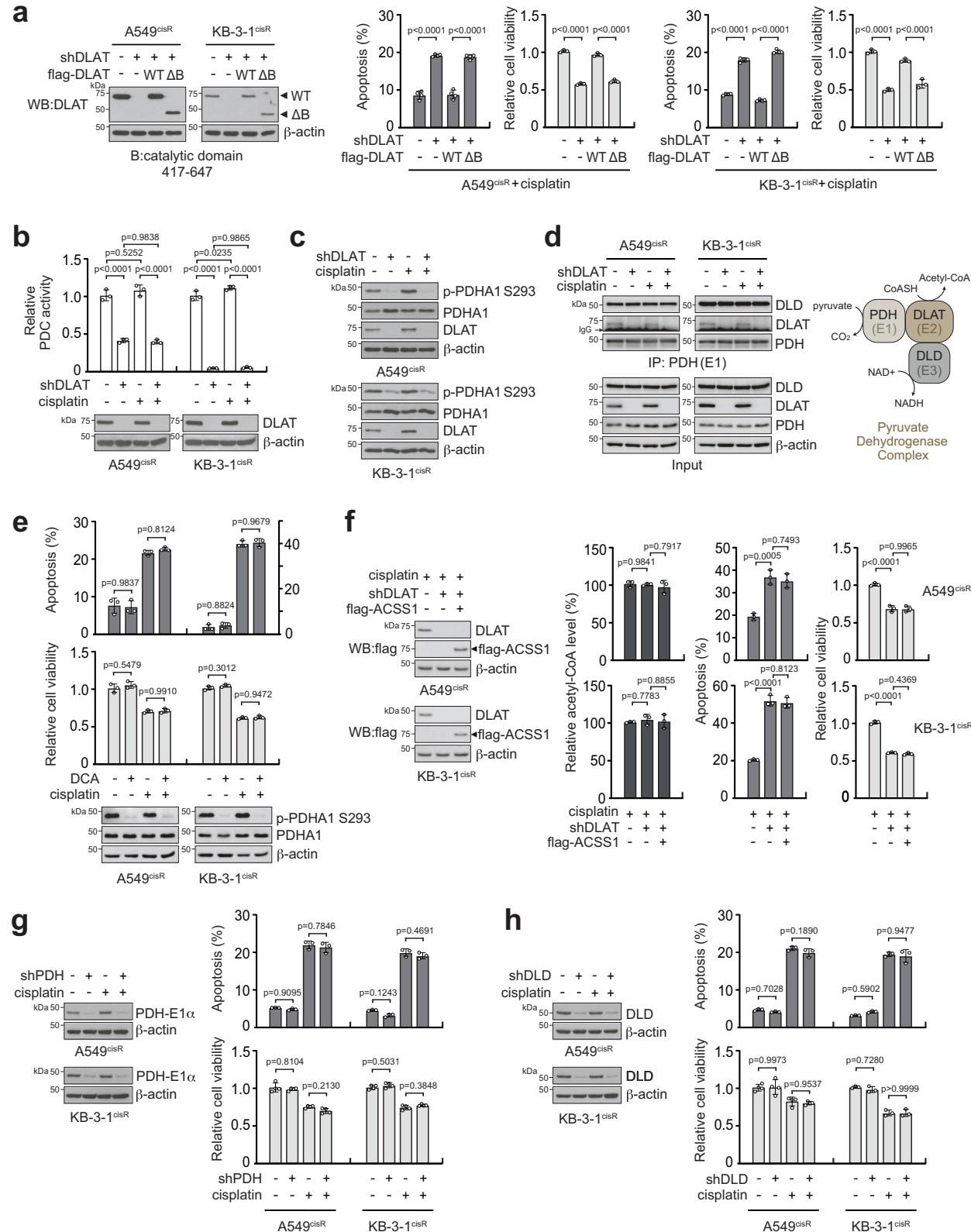

restored by 10-formyl-THF (Fig. 6c). These data suggest that the acetylation of MTHFD2 at K44 by DLAT contributes to cisplatin-resistant cell survival through a unique intermediate metabolite in folate metabolism, 10-formyl-THF. Subunits of cytochrome c oxidase, such as COX6c and COX4, are implicated in cancer metabolism and drug resistance[47,48]. Studies have shown that folate metabolism can modulate the expression of mitochondrially encoded COX subunits

through the availability of formyl donors, such as 10-formyl-THF[49]. To explore whether acetylated MTHFD2-induced 10-formyl-THF pro-motes COX subunits' expression to confer cisplatin resistance, we investigated the levels of COX in cisplatin-treated cancer cells har-boring acetyl-mimetic and -deficient mutant forms of MTHFD2. Expression of mitochondrially encoded cytochrome c oxidase II (MT-CO2), but not MT-CO1 or COXIV, was decreased in MTHFD2 K44R cells

**Fig. 2 | PDC-independent DLAT activity is needed for cisplatin-resistant cancer growth. a** Cisplatin response of cancer cells with endogenous DLAT knockdown and rescue expression of human DLAT wildtype (WT) or enzyme-inactive mutant (ΔB; catalytic domain B deleted). DLAT knockdown A549$^{cisR}$ and KB-3-1$^{cisR}$ cells expressing DLAT variants were under cisplatin-treated conditions (A549$^{cisR}$ 2 μg/ml and KB-3-1$^{cisR}$ 5 μg/ml) for 48 h. Apoptosis and cell viability were assessed by annexin V staining and ATP measurement, respectively. Total DLAT levels in cells with DLAT variants are shown by immunoblotting. **b, c** DLAT knockdown effect on PDC activity. A549$^{cisR}$ and KB-3-1$^{cisR}$ cells with DLAT knockdown were treated with cisplatin for 24 h. PDC activity was determined by monitoring NADH production from PDH reaction (**b**) and phosphorylation of PDHA1 at S293 (**c**). **d** DLAT knockdown effect on PDC assembly. Interaction between E1 (PDH), E3 (DLD), and E2

(DLAT) was determined by E1 co-immunoprecipitation in cells with or without DLAT knockdown and cisplatin treatment. **e** Effect of PDC inhibition on cisplatin sensitivity. Cells were treated with 5 mM PDK inhibitor dichloroacetate (DCA) and cisplatin for 48 h. Apoptotic cell death and cell viability were measured as described in (**a**). **f** Overexpression of ACSS1 in cells lacking DLAT. Acetyl-CoA levels were measured by LC-MS. Apoptotic cell death and cell viability were measured as described in (**a**). Effect of PDH (**g**) or DLD (**h**) knockdown on cisplatin-induced apoptotic cell death and cell viability. Data are mean ± SD from 4 independent biological replicates for apoptosis rates of (**a**) and cell viability of (**g, h** left) and 3 independent biological replicates for (**b, e, f**), cell viability of (**a, h** right), apoptosis rates of (**g, h**). *P* values were determined by one-way ANOVA for all panels. Source data are provided as a Source Data file.

and restored by the addition of 10-formyl-THF (Fig. 6d). Genetic target downregulation of MT-CO2 mimicked DLAT or MTHFD2 loss in cisplatin-resistant cancer cell survival and proliferation (Supplementary Fig. 7). Furthermore, overexpression of MT-CO2 rescued the cisplatin-resistant cell growth decreased due to the absence of MTHFD2 K44 acetylation or DLAT (Fig. 6e, f). To gain mechanistic insight into how MT-CO2 is upregulated, we genetically down-regulated four factors known to be crucial for mitochondrial gene expression and predicted to induce MT-CO2 in cells. POLRMT but not TFAM, TFB2M, or NRF1 knockdown reduced MT-CO2 levels in cells supplemented with 10-formyl-THF (Supplementary Fig. 8a). Furthermore, treatment with POLRMT inhibitor IMT1 mitigated the MT-CO2 expression induced by 10-formyl-THF (Supplementary Fig. 8b). Collectively, these data demonstrate that the acetylation of MTHFD2 by DLAT drives cisplatin resistance in part by 10-formyl-THF that promotes the expression of MT-CO2.

## DLAT-MTHFD2 signaling axis correlates with chemotherapy-resistant cancer progression in patients

To determine the clinical applicability of our findings, immunohistochemistry (IHC) staining was carried out using primary tumor samples from head and neck squamous cell carcinoma (HNSCC) and non-small cell lung cancer (NSCLC) patients who received chemotherapy-containing regimens. HNSCC patients who had no evidence of disease for over two years after the cisplatin or carboplatin-containing chemotherapy were grouped as cisplatin-sensitive (cis$^S$), and patients whose disease recurred within two years were considered cisplatin-resistant (cis$^R$). The levels of DLAT and acetyl-MTHFD2 were higher in tumors from the cisplatin-resistant group compared to a cisplatin-sensitive group of HNSCC patients (Fig. 7a, b). Moreover, DLAT and acetyl-MTHFD2 levels positively correlated in these tumors (Fig. 7c). Chemoimmunotherapy, which combines chemotherapy and immunotherapy, has become one of the standard treatments for NSCLC. Thus, we next inquired whether the levels of DLAT and acetyl-MTHFD2 align with therapy resistance in NSCLC patients receiving chemotherapy and immunotherapy. Progressive disease is defined as at least a 20% growth in the size of the tumor or spread of the tumor since the beginning of treatment. Partial response (tumor size decrease >30%) or stable disease (tumor size increase <20% or decrease <30%) was considered therapy-sensitive, while progressive disease was considered therapy-resistant. In line with the observation seen in HNSCC patient tumors, DLAT and acetyl-MTHFD2 levels were higher in the resistant group than in the sensitive group of NSCLC patients who received chemotherapy-containing regimens (Fig. 7d, e). In addition, we observed a positive correlation between DLAT and acetyl-MTHFD2 levels in these tumor specimens (Fig. 7f). MT-CO2 staining was similar to that of DLAT and acetyl-MTHFD2 in these tumors (Supplementary Fig. 9). Moreover, DLAT, acetyl-MTHFD2, and MT-CO2 levels were elevated during treatment (Supplementary Fig. 10a–d). Furthermore, post-treatment tumors from therapy-resistant patients showed significantly higher levels of DLAT, acetyl-MTHFD2, and MT-CO2 than

those from sensitive patients. In contrast, this difference was not observed in pre-treatment tumors, suggesting the potential role of DLAT signaling in acquired resistance (Supplementary Fig. 10e–g). These data collectively provide clinical validation of our findings and demonstrate a functional connection between DLAT signaling and the response of human cancers to chemotherapy.

## Identification of a decoy peptide DMp39 as a DLAT-MTHFD2 inhibitor and cisplatin sensitizer

Our finding that DLAT-MTHFD2 correlates with patient therapy response and targeting this signaling axis enhances cisplatin response indicates that DLAT-MTHFD2 could serve as a promising therapeutic target for treating human cancers that are less responsive to the chemotherapy agent cisplatin. From our perspective, there is no drug that targets the non-canonical and PDC-independent function of DLAT that is linked to MTHFD2. We, therefore, designed a decoy peptide DMp39 that mimics the K44 containing alpha helix of MTHFD2 (VISGRKLAQ-QIKQEVRQEVEEWVASGNK) and has the TAT peptide (YGRKKRRQRRR) for cell penetration, which led DMp39 mainly to the mitochondria where the target MTHFD2 is located in cancer cells (Fig. 8a and Supplementary Fig. 11). Time-dependent stability assay demonstrated that DMp39 is stable at the optimal dose of 20 μM (Fig. 8b). We assessed the efficacy of DMp39 in suspending cisplatin-resistant cancer growth both in vitro and in vivo. DMp39 specifically attenuated MTHFD2 K44 acetylation in a dose-dependent manner (Fig. 8c, d). Notably, treatment with DMp39 significantly enhanced apoptotic cell death and attenuated cell growth in combination with cisplatin in various types of cancer cells, including lung, head and neck, breast, cervical, and ovarian cancers (Fig. 8e and Supplementary Fig. 12). Next, we tested the in vivo efficacy of DMp39 in treating xenograft mouse models. For in vivo toxicity studies, 0.1 mg/kg/day of DMp39 was subcutaneously injected into the tumor area, and 5 mg/kg/day of cisplatin was administered intraperitoneally to mice every 3 days. DMp39 and cisplatin treatment had minimal organ toxicity (Fig. 8f). In addition, there were no significant histopathological property alterations between the vehicle-treated and drug-treated groups (Supplementary Fig. 13). Cy7-labeled DMp39, locally injected near the tumor lesion, effectively penetrated tumors and abolished K44 acetylation in both KB-3-1$^{cisR}$ and lung cancer patient-derived xenografts (PDX) (Fig. 8g). Consistent with genetic inhibition, while DMp39 did not alter PDC activity, it significantly reduced the level of MTHFD2 product, 10-formyl-THF, in cells and xenograft mice (Supplementary Fig. 14). Lastly, we demonstrated the efficacy of DMp39 in enhancing cisplatin response using cisplatin-resistant KB-3-1$^{cisR}$ xenografts and lung cancer PDX models. In the KB-3-1$^{cisR}$ xenograft models, treatment with cy7-labeled DMp39 in combination with cisplatin significantly reduced tumor volume compared to cisplatin or DMp39 treatment alone (Fig. 8h). The tumor volume in the combination group showed a marked decrease from day 14 onwards, reaching a statistically significant difference by day 22. Consistently, the tumor weight at the end of the study was significantly lower in the combination treatment group compared to the single-

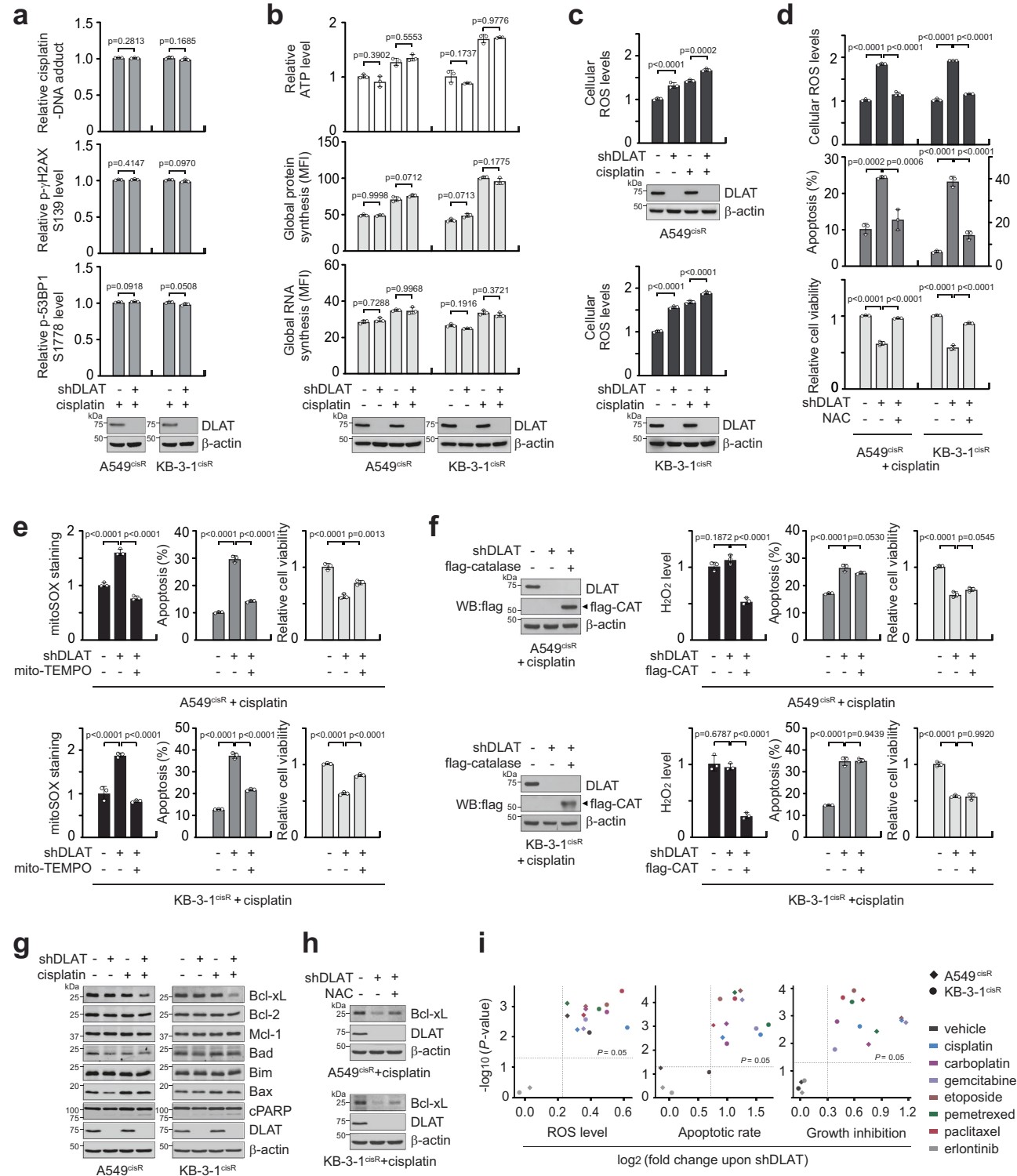

agent groups (Fig. 8i). IHC analysis of KB-3-1$^{cisR}$ xenograft tumors revealed that the combination of DMp39 and cisplatin decreased the levels of acetylated K44 MTHFD2 and MT-CO2, while the levels of DLAT remained unchanged (Fig. 8j). These results are consistent with those observed in vitro. Similarly, in the lung cancer PDX model, the combination treatment of cy7-labeled DMp39 and cisplatin led to a significant reduction in tumor volume and weight (Fig. 8k, l). In line with the observation of the IHC results from KB-3-1$^{cisR}$-derived xenograft model, IHC staining of PDX tumors demonstrated that the levels of Ac-K44 MTHFD2 and MT-CO2 were significantly decreased in the combination treatment group, while DLAT levels were unaffected (Fig. 8m). Collectively, these findings indicate that DMp39, a decoy peptide targeting DLAT-MTHFD2 signaling, has the potential to enhance the anti-tumor effect when combined with cisplatin in human cancers.

## Discussion

Acetylation, a reversible post-translational modification, is intrinsically connected with various cellular biological processes, and aberrant acetylation is viewed as one of the key features of cancer and an

**Fig. 3 | DLAT contributes to chemotherapy resistance by controlling mitochondrial ROS. a** Effect of targeting DLAT on cellular uptake of cisplatin and cisplatin-mediated DNA damage. Cells were treated with sublethal doses of cisplatin for 24 h, followed by staining with fluorescently labeled antibodies against cisplatin-DNA adducts, phospho-Histone H2A.X, and phospho-53BP1.Effect of DLAT loss and cisplatin treatment on bioenergetics, biosynthesis, and ROS levels. Cells were treated with cisplatin, and ATP, RNA, protein synthesis levels (**b**) and cellular ROS levels (**c**) were measured by luminescent assay, metabolic labeling, and DCFDA staining, respectively. **d** Cisplatin-treated cells with DLAT knockdown were treated with antioxidant NAC (0.5 mM). Cellular ROS (upper), apoptosis (middle), and cell viability (lower) were measured. Effect of mitochondria-targeted antioxidant mito-TEMPO (**e**) or catalase overexpression (**f**) on ROS levels, apoptosis, and cell viability in cells with DLAT knockdown and cisplatin. Mito-TEMPO (10 µM) or flag-tagged catalase was introduced in cisplatin-treated cells. Mitochondrial ROS and cytoplasmic hydrogen peroxide were measured by mitoSOX staining and luminescence detection, respectively. **g** Immunoblotting of apoptosis-associated proteins. Cells with or without DLAT knockdown were treated with sublethal doses of cisplatin for 24 h. **h** Effect of DLAT loss and NAC treatment on Bcl-xL expression. **i** Scatter plot of drug sensitivity in KB-3-1$^{cisR}$ and A549$^{cisR}$ representing ROS level, apoptotic rate, or growth inhibition $vs$ -log$_{10}$($P$ value). Cells were treated with sublethal doses of drugs (2 µg/ml cisplatin, 50 µM carboplatin, 10 nM gemcitabine, 10 µM etoposide, 0.5 µM pemetrexed, 3 nM paclitaxel, 1 µM erlotinib) for 48 h followed by DCFDA staining (left), annexin V staining (middle), and CellTiter-Glo viability assay (right). Fold changes were obtained by comparing the DLAT knockdown group to the control group. Data are mean ± SD from 3 independent biological replicates for (**a–f, i**). $P$ values were determined by two-tailed Student's $t$-test (**a, i**) and one-way ANOVA for the rest. Source data are provided as a Source Data file.

emerging target for cancer therapy[17]. Our finding suggests a mechanism that involves a unique mitochondrial acetyltransferase DLAT serving beyond its known role in pyruvate decarboxylation as a primary force in driving cancer cells to resist chemotherapy. DLAT induces acetylation of an essential player in one-carbon metabolism, MTHFD2, which activates MTHFD2, triggering changes in the profiles of metabolites involved. The intermediate one-carbon metabolite 10-formyl-THF was identified as a pivotal factor that renders cancer cells able to survive under cisplatin pressure. 10-formyl-THF contributes to chemoresistance in part by enhancing the mitochondrially encoded gene for MT-CO2 and alleviating ROS effects during cisplatin treatment.

The absence of DLAT did not directly alter the uptake of cisplatin or DNA damage mediated by cisplatin but enabled an accumulation of mitochondrial ROS upon cisplatin exposure, leading to enhanced apoptotic cell death. Not only cisplatin but also other chemotherapy agents, such as gemcitabine and pemetrexed, can generate oxidative stress in tumor cells, resulting in apoptotic cell death[50–52]. Aligned with this concept, DLAT loss enhanced chemotherapy-mediated mitochondrial ROS and apoptotic cell death. This implies that targeting DLAT may be not only beneficial for enhancing the response to cisplatin-based therapy but also broadly effective with other chemotherapy agents that induce metabolic redox imbalance.

Our study suggests that DLAT directly acetylates MTHFD2 at lysine 44, and removing this specific acetylation impairs the enzymatic activity of MTHFD2 to produce its metabolic product 10-formyl-THF. Eliminating this acetylation by mutation fully impairs the role of MTHFD2 in resisting cisplatin treatment in cancer cells. These data indicate that although six acetylated lysine residues exist in MTHFD2 in human cells, the acetylation at lysine 44 is the predominant event that leads to MTHFD2 activation and resistance to chemotherapy agents in cancer cell lines and patient tumor specimens we examined. Overexpression of MT-CO2 significantly, but only partially, restored the apoptotic cell death induced by loss of DLAT or MTHFD2 acetylation. This suggests that while DLAT serves as a central signal mediator, there are additional downstream effectors of DLAT-MTHFD2 beyond MT-CO2 contributing to the resistance to chemotherapy. Comprehensive multi-omics studies monitoring global transcriptional changes as well as metabolite-protein interaction proteomics further investigating 10-formyl-THF interacting proteins are warranted.

Lysine 88 of MTHFD2 is known to be deacetylated by SIRT3, and SIRT3 expression is inhibited by cisplatin, resulting in hyperacetylation of MTHFD2 in colorectal cancer[36]. However, our acetyl-proteomics study revealed that DLAT was not the acetyltransferase responsible for MTHFD2 lysine 88 acetylation, and the loss of function study showed that DLAT is not responsible for cisplatin resistance in colorectal cancer. In addition, a recent study showed that copper ions induce the DLAT gene and sensitize prostate cancer cells to docetaxel, an analog of paclitaxel[46]. There was no connection between copper ions and DLAT gene induction in the cancer cells that we studied. These data

indicate that the role of DLAT and mechanistic actions in substrate activation may be contingent on different metabolic dependencies of the cancer type. Studies reported that formyl donors, such as 10-formyl-THF, can contribute to the expression of MT-CO subunits through formylmethionyl-tRNA in Jurkat T cells[49]. We demonstrated that 10-formyl-THF, in coordination with POLRMT, induces gene expression of MT-CO2 under cisplatin pressure. This may expand the functional landscape of 10-formyl-THF that involves cell-type-specific signaling adaptations.

The standard first-line treatment option for patients with advanced lung cancer is chemoimmunotherapy, a combination of chemotherapy and PD1 inhibitors. In our study, the levels of DLAT and MTHFD2 acetylation were higher in tumors of cancer patients who were unresponsive to either chemotherapy or chemoimmunotherapy than in the group who benefited from the therapy. The difference was slightly more prominent in levels of MTHFD2 acetylation than of DLAT or MT-CO2. While MTHFD2 acetylation reflects the activity, DLAT protein level may not necessarily indicate the activation status of DLAT, which is essential for its function in chemoresistance. The clinical and molecular correlation was not as pronounced for MT-CO2 as for DLAT and MTHFD2 acetylation in NSCLC patients, suggesting that additional factors beyond DLAT-MTHFD2 may contribute to the regulation of MT-CO2 expression. The correlation between DLAT and acetyl-MTHFD2 was greater in lung cancer patients receiving chemoimmunotherapy than in head and neck cancer patients receiving platinum-based chemotherapy. It is plausible that active DLAT is more abundant in lung cancer than head and neck cancer, or the dependency of DLAT on MTHFD2 activation may differ in these two cancers. Furthermore, given an alternative contribution of MTHFD2 in anti-inflammation, DLAT-MTHFD2 may have contributed not only to the response of chemotherapy but also to immunotherapy[38,53]. Considering all this information, it appears that targeting the activity of DLAT-MTHFD2 signaling may enhance current therapy response to improve therapeutic outcomes. To enhance translational relevance, future studies should evaluate DMp39 in patient-derived xenograft and organoid models that better capture clinical heterogeneity, as well as across a broader range of tumor types.

Decoy technology has been proposed as an effective therapeutic tool against various diseases[54]. Although several MTHFD2 competitive inhibitors have been identified, due to the high degree of structural similarity between the isoenzymes, these inhibitors could potentially target the ubiquitous and essential isoenzyme MTHFD1, challenging their specificity. The peptide sequence containing lysine 44 is unique to MTHFD2, and therefore, the decoy peptide targeting this acetylation residue will likely yield specificity that other inhibitors may not have. While DMp39 exhibited robust anti-tumor efficacy in multiple preclinical models with minimal acute toxicity, further pharmacokinetic and pharmacodynamic evaluations are necessary to advance clinical translation. As with other TAT-fused or cationic peptides, rapid

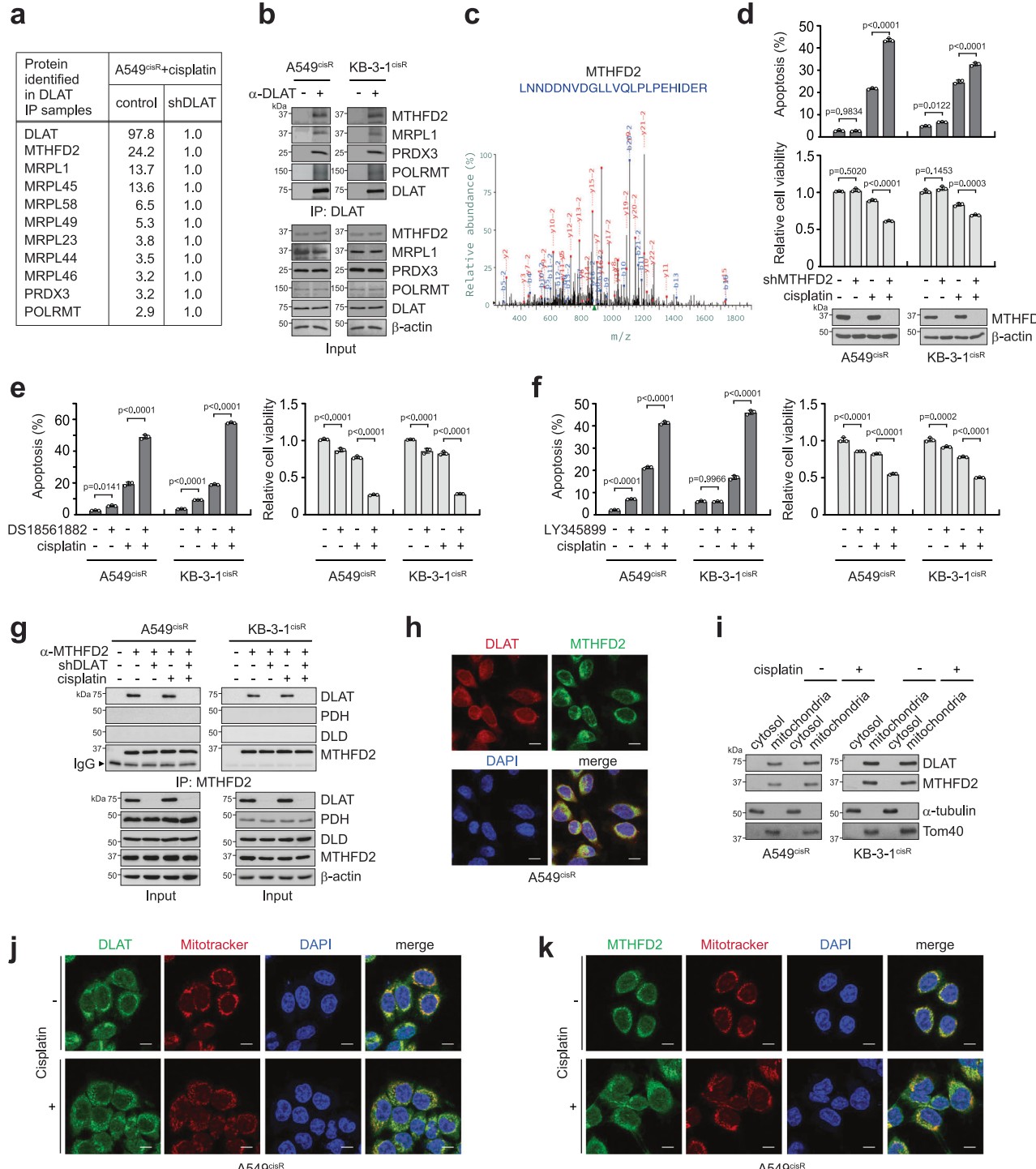

**Fig. 4 | DLAT binds to MTHFD2 in the mitochondria. a** Top 10 mitochondrial interactors of DLAT beyond PDC in A549$^{cisR}$ cells with or without DLAT knockdown. DLAT binding proteins were co-immunoprecipitated using DLAT antibody, and peptides were measured by LC-MS/MS. The normalized peak intensity value for each protein is shown. **b** Co-immunoprecipitation showing candidate interactions with DLAT. **c** Representative MS spectrum of MTHFD2 peptide fragments is shown. **d** Cisplatin-induced apoptotic cell death and cell viability in cancer cells without or with knockdown of MTHFD2. Stable knockdown cells were treated with sublethal doses of cisplatin (A549$^{cisR}$ 2 μg/ml and KB-3-1$^{cisR}$ 5 μg/ml) for 48 h. Effect of MTHFD2 inhibitors on cisplatin-mediated apoptotic cell death and cell viability. Cells were treated with either 10 μM DS18561882 (**e**) or 10 μM LY345899 (**f**) and sublethal doses of cisplatin for 48 h. **g** Endogenous interaction between MTHFD2

and DLAT, independent of other subunits of the PDC, was determined by MTHFD2 co-immunoprecipitation in cancer cells. **h** Immunofluorescence assay shows the co-localization of DLAT and MTHFD2 in A549$^{cisR}$ cells. **i** Mitochondrial localization of DLAT and MTHFD2 in A549$^{cisR}$ and KB-3-1$^{cisR}$ cells is shown by immunoblotting of mitochondria and cytosolic fractions prepared using a mitochondria isolation kit. Tom40 and α-tubulin were used as control markers for mitochondria and cytosol, respectively. Immunofluorescence staining demonstrates the localization of DLAT (**j**) and MTHFD2 (**k**) with mitochondrial marker MitoTracker in A549$^{cisR}$ cells. Scale bars represent 10 μm for (**h**, **j**, **k**). Data are mean ± SD from 3 independent biological replicates for (**d–f**). *P* values were determined by one-way ANOVA. Source data are provided as a Source Data file.

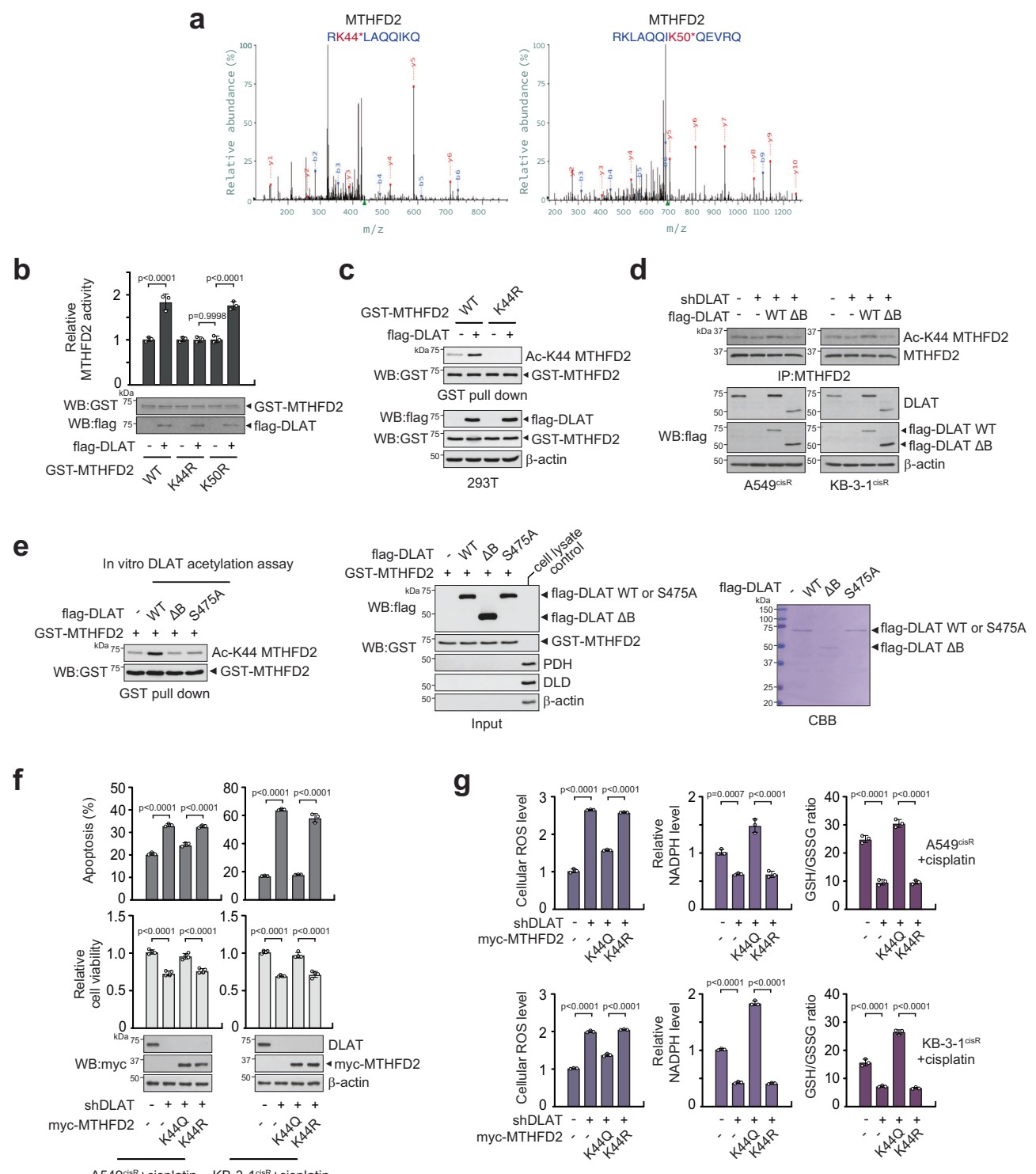

systemic clearance and short plasma half-life are anticipated, which may limit bioavailability and therapeutic durability[55]. Structural modifications such as lipidation, PEGylation, or backbone cyclization may enhance the peptide's in vivo stability and reduce proteolytic degradation[56,57]. These strategies may also mitigate potential immunogenicity commonly associated with TAT-fused or cationic peptides, thereby addressing key translational barriers to long-term therapeutic use. Given the mitochondrial localization of DMp39, it is also critical to assess long-term safety. Although our short-term dosing of DMp39 showed no acute toxicity, extended studies will be required to rule out delayed or cumulative toxicity in vital organs. These efforts

will inform rational dosing strategies and ensure safety in future translational applications. Collectively, our finding provides a compelling rationale for applying DMp39 as a targeted mitochondrial therapeutic to overcome chemotherapy resistance, and further optimization is warranted to advance DMp39 toward clinical development.

## Methods
### Constructs
The RNAi Consortium (TRC) lentiviral short hairpin RNAs (shRNAs) targeting DLAT, MTHFD2, PRDX3, PDHA1, DLD, MRPL1, POLRMT, TFAM, and TFB2M were obtained from Horizon Discovery. The shRNA

**Fig. 5 | DLAT activates MTHFD2 by acetylating at K44. a** MS spectra of acetyl-lysine peptide fragments of MTHFD2 containing K44 and K50. **b** A coupled in vitro DLAT acetylation and MTHFD2 activity assay using MTHFD2 wildtype, K44R, and K50R. In vitro acetylation assay was performed using bead-bound DLAT and purified GST-MTHFD2 variants. The acetylated MTHFD2 variants were applied to MTHFD2 activity assay using 0.2 mM tetrahydrofolate as a substrate. **c** Acetylation at K44 of MTHFD2 was assessed using a specific antibody against acetyl-K44 MTHFD2. MTHFD2 WT and K44R were applied to the in vitro DLAT acetylation assay and immunoblotting. **d** Effect of DLAT modulation on MTHFD2 K44 acetylation in A549$^{cisR}$ and KB-3-1$^{cisR}$ cells. Cells with endogenous DLAT knockdown were rescue expressed with WT or enzyme inactive ΔB DLAT. Acetylation of MTHFD2 at K44 was assessed using acetyl-K44 MTHFD2 antibody. **e** Left: In vitro DLAT

acetylation assay using GST-MTHFD2 and flag-DLAT variants, wild-type (WT), ΔB domain deletion mutant (ΔB), or S475A. Acetylation of MTHFD2 and input were assessed by immunoblotting. Right: Coomassie Brilliant Blue staining of purified flag-DLAT variants. **f, g** Effect of MTHFD2 acetylation-mimetic mutant K44Q or -deficient mutant K44R expression on apoptosis and cell viability (**f**) and cellular ROS, NADPH levels, and GSH/GSSG ratio (**g**) in DLAT knockdown cells treated with cisplatin. DLAT knockdown cells were overexpressed with myc-tagged MTHFD2 K44Q or K44R mutants, and cisplatin resistance and redox status were determined by annexin V staining and bioluminescent assays. Data are mean ± SD from 4 independent biological replicates for cell viability of (**f**) and 3 for (**b, g**) and apoptosis rates of (**f**). P values were determined by one-way ANOVA. Source data are provided as a Source Data file.

sense strands are as follows: GCAGAGGTTGAAACTGATAAA (DLAT #1), CCATACCTCATTATTACCTTT (DLAT #2), GCAGTTGAAGAAACATA-CAAT (MTHFD2), CCTAAGCCTTGATGACTTTAA (PRDX3), CGA-GAAATTCTCGCAGAGCTT (PDH), GCAGTTGAAAGAAGAGGGTAT (DLD), GCTGTATTTACAGAGAATGCA (MRPL1), GACTCCAAGGT-CAAGCAAATA (POLRMT), GTAAGTTCTTACCTTCGATTT (TFAM), and CCCAAAGCGTAGGGAATTATT (TFB2M). The MT-CO2 Human Pre-designed siRNA Set A, obtained from MedChemExpress (HY-RS08758), was used for downregulation of the MT-CO2. pJFT7-DLAT (HsCD00841815), pJFT7-MTHFD2 (HsCD00843352), pDONR223-catalase (HsCD00398454), pANT7-ACSS1 (HsCD00859262), and pANT7-MT-CO2 (HsCD00945054) were obtained from DNASU. MTHFD2 was tagged with myc at the C terminus, and DLAT, catalase, ACSS1, MT-CO2 with a flag at the C terminus (DLAT, ACSS1, MT-CO2) and N terminus (catalase) by PCR and cloned into pDEST27- or pLHCX-Gateway. A silent mutation resistant to DLAT #1 shRNA, deletion of catalytic domain B (417-647 amino acids) in DLAT, and point mutations in DLAT (S475A, M570A) and MTHFD2 (K50R, K44R, K44Q) were made by site-directed mutagenesis (New England Biolabs).

### Antibodies

Anti-pyruvate dehydrogenase E2/DLAT and MT-CO2 antibodies were obtained from Novus Biologicals (NBP2-34065 and NBP2-94364 for IHC) and Santa Cruz Biotechnology (sc-271534/B-2 for immunoblotting). Antibodies against MTHFD2 (41377/D8W9U), myc-Tag (2278/71D10), phospho-Histone gamma H2AX S139 (9718/20E3), phospho-53BP1 S1778 (2675), COX1/MT-CO1 (62101), COX2/MT-CO2 (31219), COX IV (4850/3E11), acetyl-lysine (9441), Bcl-xL (2762), Bcl2 (15071), Mcl-1 (39224/D5V5L), Bad (9268/11E3), Bim (2933/C34C5), PARP (9542), Histone H3 (4499/D1H2), LC3A/B-I/II (4108), and p62 (5114) were purchased from Cell Signaling Technology. Antibodies against PDH1-E1α (E1) (sc-377092/D-6), α-tubulin (sc-23948/B-5-1-2), Bax (sc-493), Tom40 (sc-11414/H-300), p21 (sc-397/C-19), and POLRMT/MtRPOL (sc-365082/B-1) were purchased from Santa Cruz Biotechnology. Anti-β-actin (A1978/AC-15), anti-flag (F7425), and anti-GST (G1160/GST-2) antibodies were purchased from Sigma-Aldrich. Antibodies against PRDX3 (ab128953/EPR8115), phospho-PDHA1 S293 (ab177461/EPR12200), cisplatin modified DNA (ab103261/CP9/19), and Ki-67 (ab92742/EPR3610) are from Abcam. Anti-DLD (E3) antibody (16431-1-AP) was from ProteinTech. The antibody against MRPL1 (H00065008-M02) was from Abnova. Anti-mouse IgG Alexa Fluor 488 (A11001) and anti-rabbit IgG Alexa Fluor 568 (A21069) were from Invitrogen. A custom antibody against acetyl K44 MTHFD2 was generated by PTM BIO.

### Reagents

10-Formyltetrahydrofolate (10-formyl-THF; 35763-1) was sourced from Cayman Chemical. Formate (71539), Mito-TEMPO (SML0737), N-acetyl-L-cysteine (NAC; A7250), cisplatin (1134357), etoposide (E1383), paclitaxel (580555), ATP Bioluminescent Assay Kit (FLAA) were purchased from Sigma-Aldrich. DS18561882 (HY-130251), LY345899 (HY-101943),

and IMT1 (HY-134539) were acquired from MedchemExpress. Carboplatin (S1215), gemcitabine (S1714), pemetrexed (S1135), and erlotinib (10483) were obtained from Selleckchem. Q5 Site-Directed Mutagenesis Kit (E0554) was sourced from New England BioLabs. Annexin V staining kit (556547) was purchased from BD Biosciences. ProteoExtract Protein Precipitation Kit (539180) was obtained from EMD Millipore. Mitochondria isolation kit (89874) was obtained from Pierce. MitoTracker Red CMXros (M7512) was from Invitrogen. ROS-Glo $H_2O_2$ Assay (G8820), NADP/NADPH-Glo Assay (G9081), and GSH/GSSG-Glo Assay (V6611) were procured from Promega. MtioSOX (M36008) was obtained from Invitrogen. Acetyl-CoA assay (ab87546), PDH enzyme activity assay (ab109902), EZClick Global RNA Synthesis assay (ab228561), and EZClick Global Protein Synthesis assay (ab239725) were purchased from Abcam. High-Capacity cDNA Reverse Transcription Kit (4374966) was from Applied Biosystems. Universal SYBR Green Supermix (1725122) was from Bio-Rad. D3-acetyl-CoA standard and InfinityLab Poroshell 120 EC-C18 column for LC-MS were obtained from Cayman (40458) and Agilent (699675-742), respectively. Custom peptides, DMp39 and cy7-conjugated DMp39, were synthesized by GenScript. Mouse ALT assay (ab282882) and Mouse Kidney cystatin C Assay (KE10066) were obtained from Abcam and ProteinTech, respectively.

### Cell culture

HCT116 and HT-29 colon cancer cell lines, MDA-MB231, HeLa, 293T, and KB-3-1$^{cisR}$ cells were cultured in Dulbecco Modified Eagle Medium (DMEM). The PCI-37B and FaDu cell lines was maintained in DMEM/Ham's F-12 50/50 mix medium. A549$^{cisR}$, A2780, and H1299 cells were cultured in RPMI 1640 medium. All media contained 100 U/ml penicillin and 100 μg/ml of streptomycin, supplemented with 10% fetal bovine serum. The identity of KB-3-1 and PCI-37B cells were described previously[45,58,59]. All other cell lines were from the American Type Culture Collection (ATCC). KB-3-1$^{cisR}$ and A549$^{cisR}$ cells were generated by constant exposure to cisplatin, and KB-3-1$^{cisR}$ cells are 7.21-fold, and A549$^{cisR}$ cells are 7.76-fold more resistant to cisplatin than parental cells when cisplatin IC$_{50}$ values are compared[45]. Cells with stable knockdown or overexpression of the target gene were generated by viral infection, followed by selection using puromycin (2 μg/ml) or hygromycin (300 μg/ml).

### Acetylation-related RNAi screens

Primary screening utilized the customized human acetylome-wide shRNA library. The genes related to acetylation were sorted from the PhosphoSitePlus database. KB-3-1$^{cisR}$ cells were transduced with pooled lentivirus targeting individual genes. 48 h post-infection, cells were exposed to a sublethal dose of cisplatin (5 μg/ml) in triplicates. Virus infection efficiency was assessed using puromycin (0.5 μg/ml). Cell viability was assessed after 48 h of the treatment using CellTiter-Glo Luminescent Cell Viability Assay. Candidates inducing greater than 15% cell death by gene knockdown alone and exhibiting virus infection efficacy, as evaluated by puromycin selection, were excluded. The top

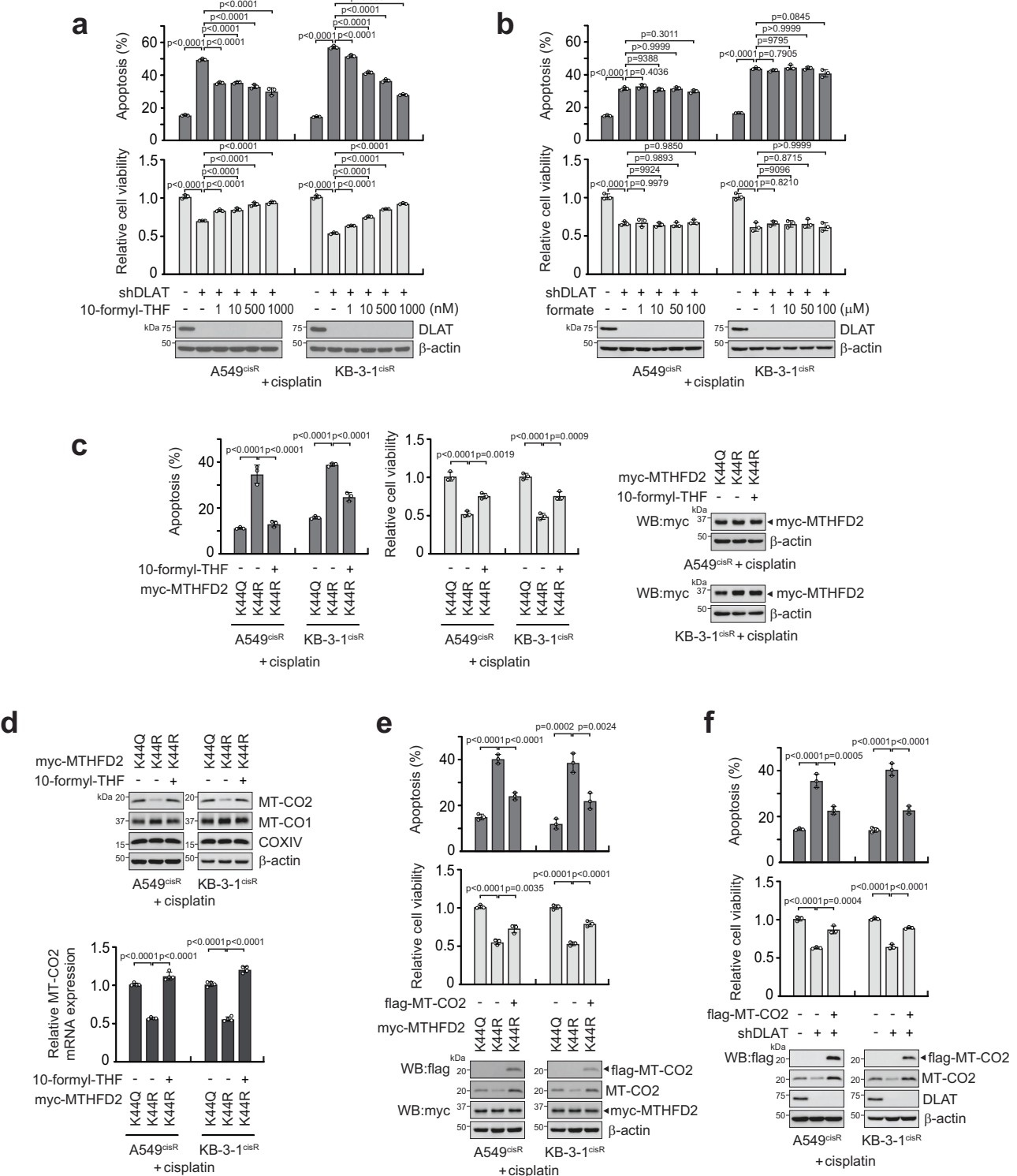

**Fig. 6 | DLAT-Acetyl-MTHFD2 contributes to cisplatin resistance through 10-formyl-THF inducing MT-CO2 expression.** Effect of supplementation with 10-formyl-THF or formate on cisplatin-resistant cell survival and apoptotic cell death in cells with DLAT knockdown. Cells cultured under sublethal doses of cisplatin were treated with 10-formyl-THF (**a**) or formate (**b**) at indicated concentrations for 48 h. Apoptosis (top) and cell viability (bottom) were determined. 10-formyl-THF rescues cisplatin-resistant cell growth (**c**) and MT-CO2 expression (**d**) in cells lacking K44 MTHFD2 acetylation. Cisplatin-treated cells expressing K44R MTHFD2 were subjected to 10 μM 10-formyl-THF for 24 h followed by annexin V staining, cell viability

assay, and assessment of MT-CO1, MT-CO2, and COXIV levels by immunoblotting and quantitative RT-PCR. Cells expressing the acetyl-mimetic mutant form of MTHFD2 K44Q were included for comparison. Effect of MT-CO2 overexpression on cisplatin-resistant cell growth in cells lacking K44 MTHFD2 acetylation (**e**) or DLAT (**f**) by K44R MTHFD2 expression or DLAT knockdown, respectively. Data are mean ± SD from 4 independent biological replicates for (**d**) and 3 for (**a**–**c**, **e**, **f**). *P* values were determined by one-way ANOVA. Source data are provided as a Source Data file.

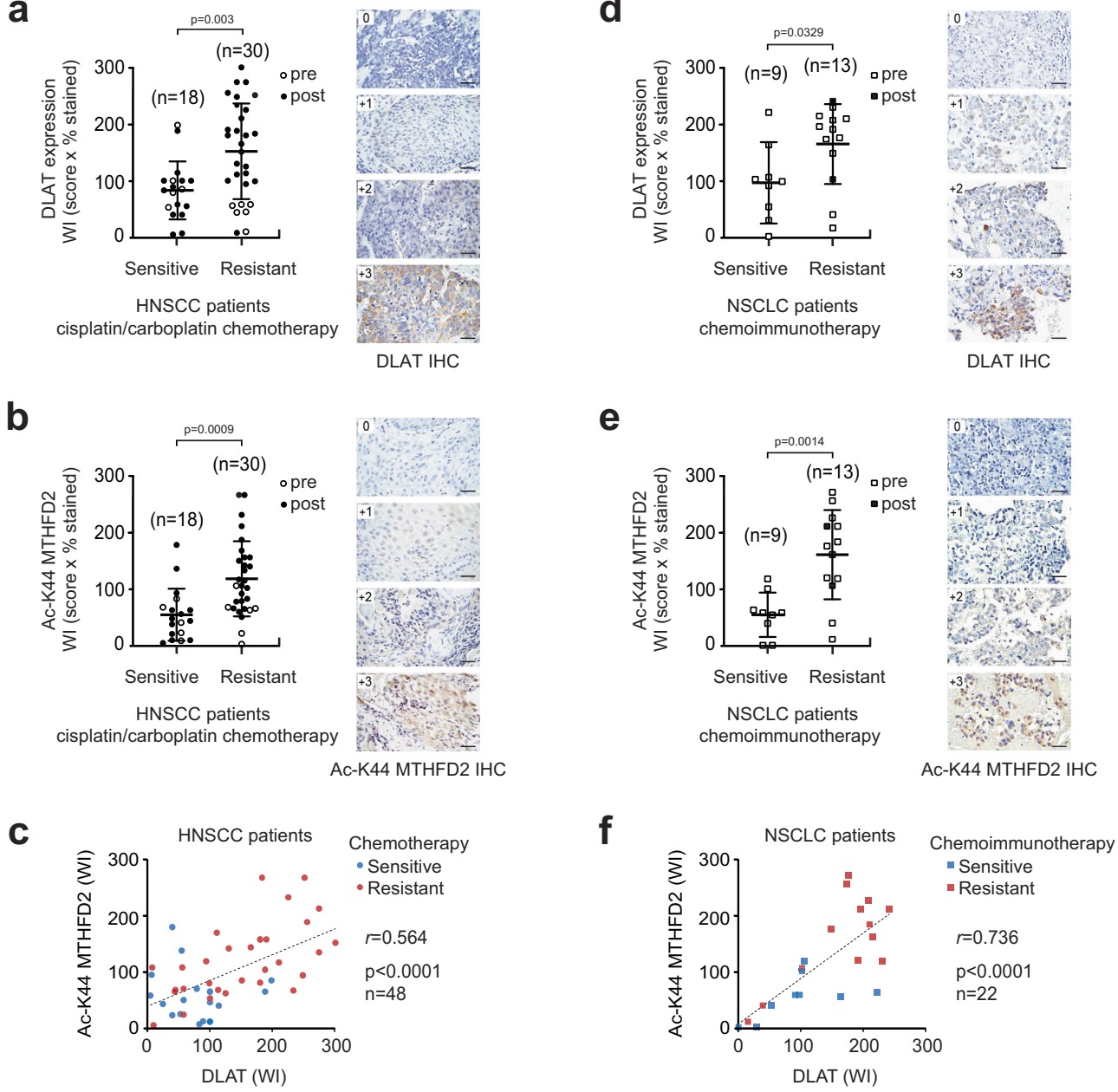

**Fig. 7 | DLAT and MTHFD2 K44 acetylation correlates with poor clinical response in cancer patients receiving chemotherapy-containing regimens.** DLAT and acetyl-MTHFD2 staining of tumors collected from head and neck squamous cell carcinoma (HNSCC) patients treated with cisplatin or carboplatin-containing regimens. NED: no evidence of disease for 2 years after treatment (therapy sensitive), recurrent: disease recurred within 2 years of treatment (therapy resistant). DLAT expression (**a**) and MTHFD2 K44 acetylation (**b**) levels in treatment-sensitive and -resistant patient tumor specimens collected after the treatment. Representative images for each staining score (0 - + 3) are shown. **c** DLAT and acetyl-K44 MTHFD2 correlation in cisplatin/carboplatin-treated HNSCC patient samples. DLAT and acetyl-MTHFD2 staining of tumors collected from non-

small cell lung carcinoma (NSCLC) patients receiving chemoimmunotherapy-containing regimens. PD: progressive disease with at least a 20% growth in the size of the tumor or spread of the tumor since the beginning of treatment (resistant); SD: stable disease or PR: partial response (sensitive). DLAT (**d**) and acetyl-MTHFD2 K44 (**e**) levels and treatment responses. **f** Correlation between DLAT and acetyl-K44 MTHFD2 in chemotherapy-treated NSCLC patient samples. Scale bars shown in (**a**, **b**) and (**d**, **e**) represent 50 μm. Results are presented as mean ± SD for (**a**, **b**, **d**, **e**). $n = 18$ (sensitive), $n = 30$ (resistant) for (**a**, **b**), $n = 48$ for (**c**), $n = 9$ (sensitive), $n = 13$ (resistant) for (**d**, **e**), $n = 22$ for (**e**). $P$ values were obtained using unpaired two-tailed Student's $t$-test (**a**, **b**, **d**, **e**) and two-tailed Pearson correlation (**c**, **f**). Source data are provided as a Source Data file.

11 candidates from the initial screen were further validated in 3 cancer cell lines, A549^cisR, KB-3-1^cisR, and H1299, as conducted in the primary screening.

**Cell viability, colony formation, cell cycle, and apoptosis assays**
Cells were seeded at a density of 5000 cells per well in 96-well plates one day prior to treatment with sub-lethal doses of cisplatin or other drugs at indicated doses for 48 h. Cell viability was determined using

the CellTiter-Glo Luminescent Cell Viability Assay. Approximately 350 cells were plated into a 35 mm dish and exposed to cisplatin for 24 h for colony formation assay. Subsequently, the cells were cultured in complete medium for an additional 10 days, and the resulting colonies were visualized by staining with crystal violet and quantified using ImageJ software. For cell cycle analysis, cells fixed with 70% ethanol were stained with RNase A (100 μg/mL) and propidium iodide (50 μg/ml) for 30 min at 37 °C, and DNA content distribution was

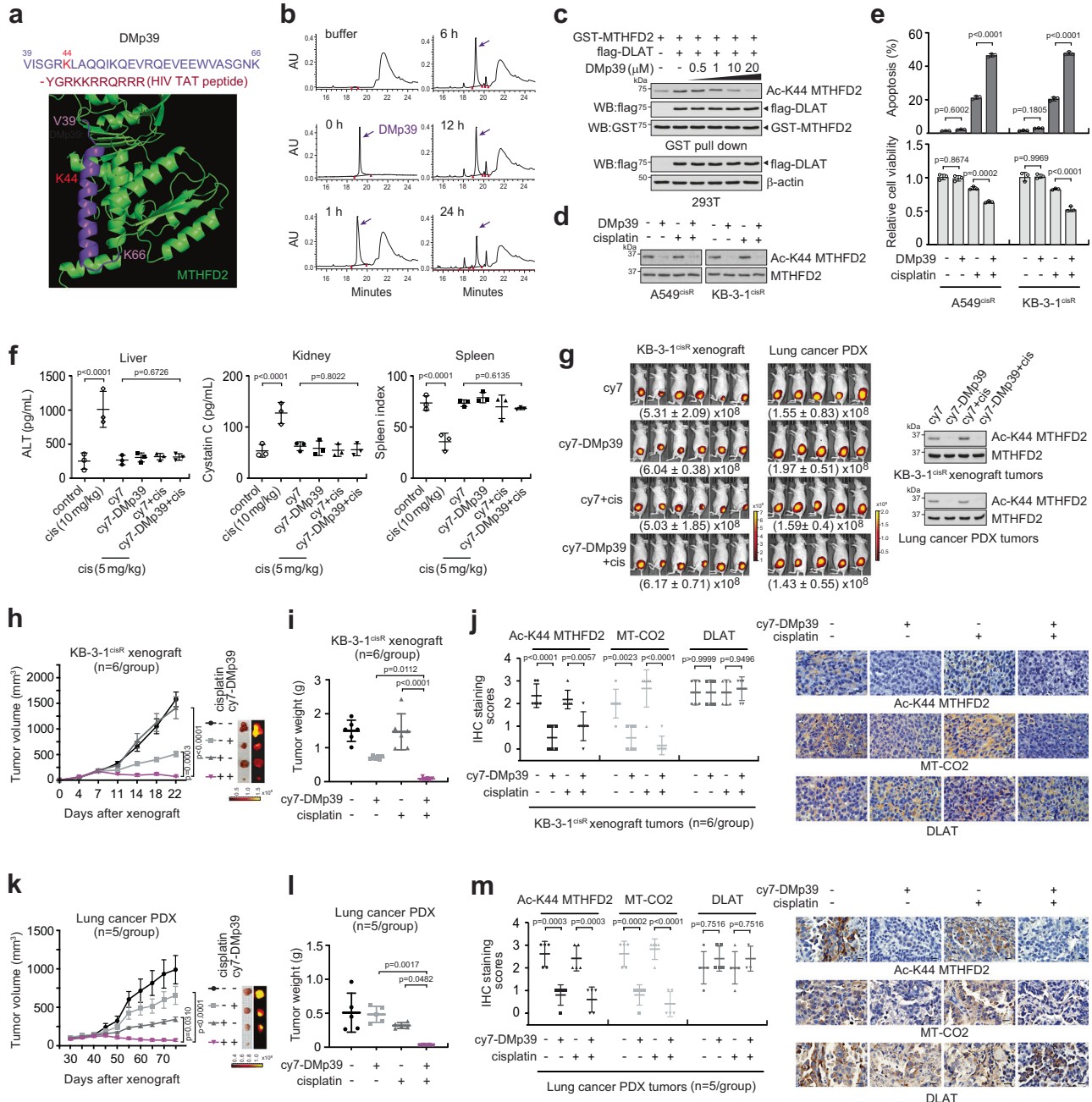

**Fig. 8 | Decoy peptide DMp39 targets DLAT-mediated MTHFD2 acetylation and increases tumor chemosensitivity in vitro and in vivo. a** The structure of MTHFD2. K44 containing α-helix in MTHFD2, is marked in purple. The amino acid sequence of DMp39 is shown at the top. **b** HPLC-based in vitro serum stability assay of DMp39. DMp39 (0.2 mg) was incubated for indicated times with pre-warmed 50% serum and applied to HPLC. The remaining DMp39 is shown by the heights of the respective elution peaks. **c** Cells were treated with the indicated concentrations of DMp39 before detection of MTHFD2 K44 acetylation. **d, e** Effect of DMp39 on cisplatin sensitivity. DMp39 (20 μM) and cisplatin (2 or 5 μg/ml) were administered for 48 h. MTHFD2 acetylation, apoptotic cell death, and cell viability were determined. **f** Organ toxicity of the cisplatin and cy7-DMp39 combination in mice. Mice were injected with 0.1 mg/kg DMp39 and 5 mg/kg cisplatin from 7 days post xenograft 2 times/week for 22 days. The positive control group was administered 10 mg/kg/day of cisplatin for 3 consecutive days. Liver, kidney, and spleen toxicity were assessed by serum alanine aminotransferase, cystatin C, and spleen index

(spleen weight (mg)/body weight (g) x 10), respectively. **g** Efficacy test of Cy7-DMp39 in vivo. Cy7 or cy7-DMp39 and cisplatin were administered as described in (**f**), and fluorescence was imaged and quantified in mice with KB-3-1^cisR^-derived xenograft tumors (**g**; left) and with lung cancer patient-derived xenograft tumors (**g**; right). MTHFD2 and Ac-K44 MTHFD2 levels in tumors were assessed by co-immunoprecipitation of MTHFD2. Effect of DMp39, cisplatin, and the combination on tumor growth (**h, k**) and weight (**i, l**) in KB-3-1^cisR^ and lung cancer PDX tumor-bearing mice. Representative images of the tumors at the endpoint are presented in (**h**) and (**k**). Ac-K44 MTHFD2, MT-CO2, and DLAT levels in both xenograft mouse tumors were assessed by IHC staining (**j, m**). Scale bars in (**h**) and (**k**) represent 10 mm and 25 μm in (**j**) and (**m**). Results are presented as mean ± SEM for (**h, k**) and mean ± SD for the rest. $n = 3$ per group for (**e, f**), $n = 6$ for (**g**) left, $n = 5$ for (**g**) right, $n = 6$ for (**h–j**), and $n = 5$ for (**k–m**). $P$ values were obtained by one-way ANOVA. Source data are provided as a Source Data file.

measured using flow cytometry (BD Symphony A3) and analyzed with FlowJo software (v10.9). Apoptotic cell death was assessed via FITC-conjugated annexin V and propidium iodide staining.

## Cellular metabolism assays

Cellular metabolic states were examined through various assays. To gauge the levels of total intracellular or mitochondrial reactive oxygen species (ROS), cells were stained with CM-H2DCFDA or mitoSOX, respectively. Intracellular $H_2O_2$, NADPH, acetyl-CoA, and GSH/GSSG ratios were quantified using commercial assay kits as listed in the Reagents. Intracellular ATP levels were determined utilizing the ATP bioluminescent somatic cell assay and normalized to the total protein concentrations. The de novo synthesized RNA and proteins were assessed in cancer cells using commercial assays following the manufacturer's protocols. In brief, O-propargyl-puromycin or uridine analog, 5-EU, was applied to stop translation or modify RNA. Polypeptides or modified RNA were measured using fluorescent azide with Ex/Em at 494/521 nm. PDC activity was determined by measuring the reduction of NAD+ to NADH during the conversion of pyruvate into acetyl-CoA.

## Quantification of acetyl-CoA and 10-formyl-THF by LC-MS

Targeted LC-MS analysis was performed to quantify acetyl-CoA from cultured cells ($n = 3$/group) and 10-formyl-THF from tumor tissues ($n = 6$/group) and cultured cells ($n = 3$/group). Cell pellets were suspended in 2.5% 5-sulfosalicylic acid spiked with 10 ng of D3-acetyl-CoA standard (Cayman) per sample. Samples were centrifuged at 17,000 x $g$ at 4 °C for 5 min, and the supernatants were subjected to LC-MS analysis using a 1260 Infinity II HPLC coupled to an Agilent 6150 single quadrupole LC/MS. In brief, 10 μL of each sample was injected onto an Agilent InfinityLab Poroshell 120 EC-C18 column (Agilent). Temperatures for the auto-sampler were set at 4 °C and the column compartment at 40 °C, respectively. The mobile phase was composed of solvent A (50 mM formic acid in $H_2O$, adjusted to pH 8.2 with ammonium hydroxide) and solvent B (100% methanol). The chromatographic gradient was run at a flow rate of 0.3 ml/min as follows: 0-1 min: hold at 20% B; 1–11 min: linear gradient from 20% to 100% B; 11–12 min: hold at 100% B; 12–13 min: linear gradient from 100% to 0% B; 13–20 min: hold at 100% B. The mass spectrometer was operated in selected ion monitoring (SIM) mode with a capillary voltage of 3.5 kV and a nozzle voltage of 2 kV. The sheath gas was maintained at 250 °C with a flow rate of 10 L/min, and the drying gas at 300 °C and 5 L/min. The retention time and position of the peaks were confirmed using pure standard compounds in $H_2O$. The peak area/height was quantified using Agilent OpenLAB CDS ChemStation Edition (Rev C.01.10 (201)). The mass to charge ratio (m/z) of the following ions was detected: acetyl-CoA (m/z 810.6), D3-acetyl-CoA (m/z 813.6), 10-formyl-THF (m/z 472.4 (quantifying), 456.4 (qualifying)). D3-acetyl-CoA was used as an internal standard for absolute quantification of acetyl-CoA, and 10-formyl-THF was used for relative quantification.

## In vitro acetylation assay

Flag-tagged DLAT enriched by flag M2 beads was used as an acetyl-transferase source. Recombinant MTHFD2 or purified GST-MTHFD2 variants were used as a substrate and mixed with Flag-tagged DLAT at a 1:1 molar ratio in HAT buffer (40 mM Tris-HCl pH 8.0, 75 mM potassium chloride, and 10 μM acetyl CoA) in a total volume of 25 μl. The reaction mixture was incubated at 30 °C for 45 min. Reactions were terminated by the addition of SDS-PAGE sample buffer followed by heating at 95 °C for 5 min. The acetylation status of MTHFD2 was assessed by immunoblotting using either pan-acetylation-specific antibody or acetyl-K44 MTHFD2 antibody.

## MTHFD2 activity assay

Recombinant or bead-bound MTHFD2 was incubated with 0.2 mM tetrahydrofolate (dissolved in 0.1 M phosphate buffer) in reaction buffer (25 mM MOPS (pH 7.3), 5 mM $K_3PO_4$, 2.5 mM formaldehyde, 36 mM β-mercaptoethanol, 5 mM $MgCl_2$, and 0.6 mM NAD+) for 10 min at 30 °C in a total reaction volume of 100 μl. Enzymatic activity was assessed by measuring the increase in absorbance at 340 nm, which reflects the reduction of NAD+ to NADH.

## Pyruvate dehydrogenase (PDH) activity assay

To estimate the activity of the pyruvate dehydrogenase complex (PDC), PDH (E1 subunit) activity was measured using the Pyruvate Dehydrogenase Enzyme Activity Microplate Assay Kit (Abcam) according to the manufacturer's instructions. Briefly, A549[cisR] and KB-3-1[cisR] cells with or without DLAT knockdown were harvested, washed with PBS, and lysed using the supplied detergent solution. After quantifying protein concentration, lysates were diluted to 10 mg/ml in 1 x buffer, and 200 μl of assay solution was added to each well, and absorbance at 450 nm was measured kinetically every 30 s for 30 min at room temperature using a microplate reader. PDH activity, which reflects the functional status of the E1 component, was calculated as the rate of absorbance increase used as a surrogate readout of PDC activity.

## Proteomics analysis

DLAT-interacting proteins were identified through DLAT co-immuno-precipitation, followed by microcapillary LC-MS/MS analysis. In brief, A549[cisR] cells with or without DLAT knockdown were lysed in 1% NP-40 lysis buffer (50 mM Tris-HCl (pH 7.4), 150 mM NaCl) supplemented with protease inhibitor cocktail (Millipore). After centrifugation, 2 mg of total protein was incubated with 10 μg of anti-DLAT antibody for 2 h at 4 °C, followed by an additional 2 h incubation with Protein G Sepharose 4 Fast Flow (Cytiva). The bead-bound proteins were washed three times with lysis buffer, and extracted and denatured using 0.12 M Tris (pH 6.8), 3.3% SDS, 10% glycerol, and 3.1% dithiothreitol. The proteins were precipitated using a protein precipitation kit (Millipore) and analyzed by the Taplin Mass Spectrometry Facility at Harvard Medical School.

## Immunofluorescence microscopy analysis

Cells cultured on glass coverslips were treated with 100 nM Mito-Tracker dye for 30 min at 37 °C, followed by fixation and permeabilization using PHEMO buffer (68 mM PIPES, 25 mM HEPES, 3 mM $MgCl_2$, 15 mM EGTA, 3.7% formaldehyde, 0.05% glutaraldehyde, 0.5% Triton X-100) for 10 minutes. Cells were probed with the primary antibody specific for DLAT or MTHFD2 at a 1:100 dilution and Alexa Fluor 488 or 568 conjugated secondary antibody at a 1:1000 dilution in PBS containing 5% goat serum. Cells were mounted using an antifade mounting solution with DAPI and analyzed using a Leica SP8 confocal microscope.

## Immunohistochemistry staining

Formalin-fixed and paraffin-embedded tumor tissue specimens from HNSCC and NSCLC patients were obtained from the Winship Cancer Tissue and Pathology Shared Resources. The Institutional Review Board (IRB) of Emory University approved using human specimens (IRB00003208 - HNSCC, IRB00098377 - NSCLC). Clinical samples were collected with informed consent according to the Health Insurance Portability and Accountability Act. Patients' clinical information was obtained through the Winship Data and Technology Applications Shared Resource. Tumors from HNSCC or NSCLC patients who received chemotherapy-containing regimens were used. Immunohistochemistry (IHC) staining of DLAT, acetyl-K44 MTHFD2, MT-CO2, and Ki-67 was performed based on the established protocol[60]. Antigen retrieval was conducted in Tris-EDTA (pH 9.0) for acetyl-K44 MTHFD2 and MT-CO2 and sodium citrate (pH 6.0) for the remaining antigens. Antibodies against acetyl-K44 MTHFD2 were probed at 1:2000 and DLAT and Ki-67 at 1:1000, and MT-CO2 at 1:250 dilutions. Assessment

of positive DLAT, MT-CO2, and acetyl-K44 MTHFD2 staining intensity within tumor cells was performed using a scoring system ranging from 0 to 3+ based on IHC signal intensity.

## Animal studies

All animal procedures were conducted based on the protocol (DAR-PROTO-201700885) approved by the Emory University Institutional Animal Care and Use Committee. Female athymic nu/nu mice (Hsd:Athymic Nude-Foxn1nu, 6-week-old; Envigo) were housed under specific pathogen-free (SPF) conditions in the Emory University animal facility. Mice were maintained under a 12 h light/12 h dark cycle, at an ambient temperature of $22 \pm 2\,^{\circ}C$ and relative humidity of 40–60%. A total of 5–8 mice per group were used for in vivo experiments.

Mice were subcutaneously injected into the flank with $1 \times 10^6$ KB-3-1$^{cisR}$ cells with or without DLAT knockdown. Tumor-bearing mice were intraperitoneally administered cisplatin (5 mg/kg) twice weekly once tumors reached approximately 100 mm$^3$ in size. To examine the effect of DMp39 on cisplatin response in vivo, KB-3-1$^{cisR}$ cells or lung cancer patient tumors (Jackson Laboratory, TM00219) were subcutaneously implanted into athymic mice. Once the tumors reached approximately 100–150 mm$^3$ in size, the mice were randomly divided into four groups and intraperitoneally administered cisplatin (5 mg/kg) twice a week, and cy7-conjugated DMp39 (0.1 mg/kg) or cy7 control was subcutaneously injected into the tumor site every other day. The tumor volume of KB-3-1$^{cisR}$ or PDX tumors was monitored by blinded measurements of two perpendicular diameters and calculated using the formula $4\pi/3 \times (width/2)^2 \times (length/2)$. Tumor size did not exceed the IACUC-approved limit of 2.0 cm in diameter. Tumors were collected at the experimental endpoint, and tumor proliferation was assessed using Ki-67 IHC staining. The localization, penetration, and efficacy of DMp39 were monitored by assessing the acetyl-K44 levels in tumors and fluorescence imaging analysis using an IVIS imaging system. Drug-induced organ toxicities, including splenomegaly, liver damage, and kidney injury, were monitored by assessing spleen index, alanine transaminase activity, and cystatin C levels in plasma.

## Statistics and reproducibility

Graphical presentations and statistical analyses were performed using GraphPad Prism 10.0. A statistical method was not employed to set the sample size. The data presented are from one of the multiple experiments performed. Data with error bars indicate the mean ± standard deviation (SD), except for Figs. 1g, 8h, k, which indicate the mean ± standard error of the mean (SEM). Statistical significance was assessed using an unpaired two-tailed Student's t-test for Figs. 3a, i, 7a, b, d, e, paired two-tailed Student's t-test for Fig. 1c and Supplementary Fig. 10a, two-way ANOVA for tumor volume, and one-way ANOVA for all others. Normal distribution and homogeneity of variances are assumed for the statistical tests used. The variation within each group was measured using the standard deviation for statistical analysis.

## Reporting summary

Further information on research design is available in the Nature Portfolio Reporting Summary linked to this article.

## Data availability

The proteomics data generated in this study have been deposited in the PRIDE repository via the ProteomeXchange Consortium under accession code PXD064757. Publicly available datasets from KMplot (https://kmplot.com/analysis/), AlphaFold (https://alphafold.ebi.ac.uk/entry/P13995), PhosphoSitePlus (https://www.phosphosite.org/homeAction.action) were also used in this study. The remaining data are available within the Article, Supplementary Information, or Source Data file. Source data are provided with this paper, which includes uncropped and unprocessed scans of key blots. Source data are provided with this paper.

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

## Acknowledgements

We thank Dr. Anthea Hammond for her editorial assistance. We thank the Taplin Biological Mass Spectrometry Facility and Ross Tomaino for conducting proteomics studies. Part of the metabolomics studies were performed by the Penn Metabolomics Core. This work was supported in

part by NIH/NCI grants R01 CA175316 (S.K.), R01 CA266613 (S.K.), R01 CA 287782 (S.K.), R21 CA277103 (S.K.), R21 CA274620 (A.A.I.), P01 CA257906 (H.F.), P50 CA217691 (S.R.), Mary Kay Ash Foundation Cancer Research Award (A.A.I) and the Emory University Integrated Cellular Imaging Microscopy, Cancer Animal Models, Cancer Tissue and Pathology, and Data and Technology Applications Cores of the Winship Cancer Institute comprehensive cancer center grant, P30 CA138292.

## Author contributions

D.M.S., N.F.S., H.F. and S.S.R. provided clinical information for the patient specimen data analyses. K.R.M. performed a histopathological study. C.-K.Q. provided conceptual advice. Jaehyun K. supported the peptide decoy study. S.W. and A.A.I. performed structural analyses. T.H. performed targeted metabolomics. JiHoon K. and K.E. conducted in vivo experiments. J.S.H., H.E.B., V.A. and S.S. performed all other experiments. J.S.H. and S.K. designed the study and wrote the paper.

## Competing interests

The authors declare no competing interests.
