## [Transparent Peer Review file · Nature Communications]

Non-canonical dihydrolipoyl transacetylase promotes chemotherapy resistance via mitochondrial tetrahydrofolate signaling

Corresponding Author: Dr Sumin Kang

Version 0:

Reviewer comments:

Reviewer #1

(Remarks to the Author)

In this manuscript, Hwang et al. present novel insights into the roles of the mitochondrial enzyme dihydrolipoyl transacetylase (DLAT) in chemotherapy resistance, specifically resistance to cisplatin. Mechanistically, the authors demonstrate that DLAT acetylates MTHFD2 at lysine 44, impacting mitochondrial folate metabolism and promoting the production of 10-formyl-tetrahydrofolate (THF). This, in turn, induces COX-2 expression independently of pyruvate dehydrogenase complex (PDC) activity. Additionally, the study highlights that DLAT modulates cell death via elevated ROS levels. In tumor specimens from chemotherapy-resistant patients, both DLAT and acetyl-MTHFD2 were found to be elevated compared to chemotherapy-sensitive cases. A particularly novel aspect of the study is the development of a decoy peptide, DMp39, which blocks DLAT-mediated MTHFD2 acetylation. In combination with cisplatin, DMp39 significantly reduced tumor growth in mouse models, suggesting potential clinical application.

The work provides valuable contributions to the understanding of chemotherapy resistance mechanisms, building on prior research linking mitochondrial function with tumor drug resistance through metabolic reprogramming and antioxidant stress responses. While previous studies have emphasized the role of MTHFD2 in purine synthesis and mitochondrial regulation, this study uniquely expands our knowledge by introducing the DLAT-MTHFD2 axis as a novel mechanism of resistance. This discovery may open new therapeutic avenues for targeting mitochondrial metabolism to overcome chemotherapy resistance. Overall, the study is comprehensive and well-supported by experimental data. The findings have translational potential, particularly the therapeutic combination approach involving DMp39.

Concerns and Suggestions for Improvement:

- Page 9-10, Fig. 6d: The authors examine whether acetylated MTHFD2-induced 10-formyl-THF promotes mitochondrial protein expression, focusing specifically on COX. The manuscript would benefit from a clearer rationale for this focus, perhaps referencing existing studies that implicate COX-2 in cisplatin resistance. This context would help justify why COX-2 was selected for investigation.

- Page 7, Fig. 3: The role of mitochondrial ROS in chemotherapy resistance is complex. While the manuscript shows that DLAT influences ROS levels, it would benefit from more discussion on how ROS accumulation affects cellular processes beyond apoptosis.

- The data on DMp39 are promising (Fig. 8). However, the long-term stability and biodistribution of the peptide in vivo could be discussed in more depth. Furthermore, testing DMp39 in additional cancer types and combination regimens, would strengthen the therapeutic claims and highlight its potential clinical relevance.

-Page 10-11, Fig. 7: The manuscript presents both pre- and post-treatment levels of DLAT-MTHFD2. It would be beneficial to clarify whether baseline levels or changes post-treatment are more critical for predicting cisplatin resistance. Additionally, the impact of cisplatin treatment on DLAT-MTHFD2 expression levels could also be explored.

-The study shows that DLAT is not involved in cisplatin resistance in colon cancer (Supplementary Fig. 2), despite its

significant effects in lung cancer models. The authors could explore or provide more context on how the DLAT-MTHFD2 interaction is regulated across different cancer types. A deeper exploration into why DLAT is ineffective in colon cancer could enhance understanding of its specificity.

-The manuscript suggests that DLAT is associated with post-target resistance to cisplatin. It would strengthen the paper if the authors could further explore and discuss post-target-related mechanisms, for example involving apoptosis-related proteins such as BCL-2 family members, or oxidative stress markers like Nrf2 and GSH.

- The mechanism by which DLAT selectively acetylates MTHFD2 over other mitochondrial proteins remains somewhat unclear. Providing additional analysis would further strengthen this key mechanistic insight.

Reviewer #2

(Remarks to the Author)

The manuscript by Hwang et al investigates mechanisms for cisplatin resistance by screening acetylation associated proteins using RNA interference in cisplatin resistant cell lines. They identify the pyruvate dehydrogenase complex subunit dihydrolipoyl transacetylase (DLAT) and its interaction with and acetylation of methylenetetrahydrofolate dehydrogenase 2 (MTHFD2) as a mediator of cisplatin resistance. The authors show that DLAT promotes acetylation of MTHFD2 at K44, which increases its enzymatic activity leading to production of 10-formyl-THF. This function is beyond DLAT known enzymatic function in the PDC, which is to transfer an acetyl group to generate acetyl-CoA, and is potentially a highly significant finding. The authors show that MTHFD2 acetylation is linked to intracellular ROS via 10-formyl-THF and COX2 induction, although the mechanism linking them is unclear. The authors go on to generate a decoy peptide based on MTHFD2 to inhibit DLAT interaction and find efficacy in cisplatin sensitisation of a resistant xenograft tumour model in mice. The manuscript is well-written, with the strength of replicating experiments across multiple cell lines. The findings are novel and exciting, though some mechanistic gaps (outlined below) hinder full evaluation of their significance.

Comments:

1. The mechanism and metabolic implications of DLAT-MTHFD2 interaction is unclear;
 - a. In Figure 2F acetyl-CoA was added to cell media with the idea that this supplemented intracellular acetyl-CoA (measured by ELISA assay). This is not a valid experiment since acetyl-CoA is an intracellular metabolite and is not directly transported across the plasma membrane and acetyl-CoA measurements by ELISA assay are known to produce erroneous results (PMID: 37398224). Moreover, mitochondrial acetyl-CoA is separated from cytosolic acetyl-CoA (PMID: 36658431), so even if extracellular acetyl-CoA could indirectly supplement intracellular acetyl-CoA, there is no indication that mitochondrial acetyl-CoA is effected. The conclusion drawn from this experiment that effects of DLAT knockdown are unrelated to [acetyl-CoA] is not substantiated.
 - b. The catalytic site deletion and mutation experiments (Fig 5F, Fig 8A) suggest that the lysine acetyltransferase mechanism is similar to acetyl transfer to CoASH. Is this activity sustained in the absence of the PDH and DLD subunits? The in vitro experiments (Fig 5A-E) suggest this subunit acts alone but PDH and DLD (and potentially other enzymes) may be associated and should be tested for. Direct comparison of WT and enzymatic mutated DLAT in vitro would clarify if the acetyl transferase activity can be directly attributable to DLAT rather than other (non PDC) associated enzymes.
2. Unanswered questions on the mechanism of action and specificity of the Dmp39 peptide include: Is Dmp39 acetylated as a competitive substrate for DLAT? Does Dmp39 affect the acetyl-CoA generation activity of the PDC? Does the peptide affect 10-formyl-THF in vivo, as per the proposed mechanism? How does the peptide enter the mitochondria? It is surprising that toxicity markers do not respond to cisplatin (Figure 8E). How can these markers be appropriate measures of (lack of) toxicity of the peptide if they are unchanged by cisplatin treatment?
3. There is no indication of how 10-formyl-THF is linked to COX2 induction. Metabolomic data would be useful to understand the specificity of the 10-formyl-THF response, and potential downstream effects. Figure 7 examines the association of DLAT and MTHFD2 with cisplatin resistance, additional analysis of COX2 and relationship to DLAT and MTHFD2 would be useful.

Minor comments:

4. Figure 2E: evidence that DCA is effective should be provided i.e. phospho PDH blots, metabolic tracing.
5. Figure 4F: MTHFD2 IP: blotting of other PDC subunits is required to understand the specificity of the interaction with DLAT. A negative control would help to evaluate the IP.
6. Schematic illustrations would be beneficial to illustrate PDC complex with each subunit labelled (Figure 2) and acetylation and mutation and sites (Figure 5).
7. Some panels are too small to read e.g Fig 4A, 5C chromatograms and Fig. 8A model.

Reviewer #3

(Remarks to the Author)

This manuscript describes the non-canonical role played by mitochondrial dihydrolipoyl transacetylase (DLAT) in acetylation of methylenetetrahydrofolate dehydrogenase 2 (MTHFD2) at lysine 44, promoting resistance to chemotherapy, such as cisplatin. The authors took logical steps to identify DLAT as one of the key players involved in acquisition of resistance to cisplatin, elucidated downstream pathway involving MTHFD2 and COX2, and demonstrated potential therapeutic strategy using a decoy peptide.

The manuscript is well-written with clear logic and seamless flow. Experimental methods were provided in detail and the data were clearly presented with logical interpretations. The overall conclusions of the manuscript are supported by the extensive data in the results section. The work presented in the manuscript is of significance to the field of anticancer research as the newly identified mechanism of drug resistance has important therapeutic implications. The only suggestion is to assess the synergistic effects of cisplatin and a COX2 inhibitor (either in vitro or in vivo) given that very selective and potent COX2 inhibitors are available.

Version 1:

Reviewer comments:

Reviewer #1

(Remarks to the Author)

The authors have made thoughtful and thorough revisions to the manuscript. This study addresses an important question in cancer biology by uncovering a novel mechanism of chemotherapy resistance involving DLAT-mediated acetylation of MTHFD2 and its downstream impact on mitochondrial folate metabolism. The additional data notably strengthen the scientific rigor of the work and satisfactorily address the majority of concerns raised during the previous review cycle.

Below are a few minor comments and suggestions:

-The newly included in vivo data on tumor-selective accumulation and stability of Cy7-DMp39 enhance the translational relevance of the study. Additionally, the use of multiple cancer models to validate DMp39 efficacy provides encouraging breadth. It may be valuable to briefly discuss the potential for future testing in resistant, patient-derived models across a wider range of tumor lineages.

-While the decoy peptide DMp39 demonstrates promising activity in preclinical models, a more detailed discussion of its pharmacokinetics and pharmacodynamics (e.g., half-life, clearance, bioavailability) would help contextualize its clinical potential and inform prospective dosing strategies. Similarly, although minimal toxicity has been observed, data from longer-term toxicity studies (e.g., over 4–6 weeks) would further support the safety profile, especially in light of the peptide's mitochondrial localization.

-Figure 8g would be strengthened by the inclusion of quantitative analysis to accompany the representative images.

-Given the reliance on TAT-based delivery for DMp39, it would be helpful to include a brief comment on potential immunogenicity or other translational barriers associated with peptide-based therapeutics.

Reviewer #2

(Remarks to the Author)

I congratulate the authors on a thorough and exciting study, with the identified links between findings consolidated in the revised manuscript. The authors identified pyruvate dehydrogenase complex (PDC) subunit dihydrolipoyl transacetylase (DLAT) and its interaction with, and acetylation of methylenetetrahydrofolate dehydrogenase 2 (MTHFD2) as a mediator of cisplatin resistance. This work contributes novel findings on the function of the core metabolic enzyme, DLAT, separate to its known role in the generation of acetyl-CoA in the PDC. The authors show that DLAT directly promotes acetylation of MTHFD2 at K44, which increases its enzymatic activity leading to production of 10-formyl-THF. Downstream, 10-formyl-THF production modulates expression of mitochondrially encoded cytochrome c oxidase II (MT-CO2) via mitochondrial RNA Polymerase (POLMRT), managing mitochondrial ROS and expression of antiapoptotic Bcl-xL. This work advances understanding of mechanisms of cisplatin resistance in cancer, and leverages novel mechanistic insight to develop an innovative treatment modality through direct targeting of MTHFD2 acetylation by DLAT using a decoy peptide. This work opens new avenues for cancer treatment, and understanding of metabolic coordination.

The revised manuscript is significantly improved, and my previous concerns were thoroughly addressed. In particular, the in vitro acetylation assay (Supp. Fig. 5b) clearly demonstrates that wild-type (WT), but not mutant, DLAT efficiently catalyzes the acetylation of MTHFD2 at K44. I suggest that Supp. Fig. 5b replace the current main Fig. 5a, as it represents a stronger experiment with appropriate controls and effectively shows that acetylation is not dependent on the presence of acetyl-CoA.

Minor comments/tips:

More detail in methods are required for in vitro acetylation assay, MTHFD2 activity assay, and proteomics assay which are missing details on cell lysis and immunoprecipitation conditions or appropriate references to such details, and for PDC

activity assay.

Figure 3h stroke missing from +

Figure 4a samples typo

~Sophie Trefely

Reviewer #3

(Remarks to the Author)

The reviewer's comments have been addressed satisfactorily in the revised manuscript, and I have no further comments.

Reviewer #4

(Remarks to the Author)

Response to Review

We greatly appreciate the thorough reviews of our manuscript NCOMMS-24-55974. We have performed extensive studies to address the insightful and constructive comments from the Reviewers, as detailed below. We believe that the manuscript has been further strengthened as a consequence of this revision. **Changes made in the manuscript are marked in red color** in the Article and Supplementary Information uploaded together with this Response to Review.

REVIEWER 1

“This discovery may open new therapeutic avenues for targeting mitochondrial metabolism to overcome chemotherapy resistance. Overall, the study is comprehensive and well-supported by experimental data. The findings have translational potential, particularly the therapeutic combination approach involving DMp39.”

1. *“Page 9-10, Fig. 6d: The authors examine whether acetylated MTHFD2-induced 10-formyl-THF promotes mitochondrial protein expression, focusing specifically on COX. The manuscript would benefit from a clearer rationale for this focus, perhaps referencing existing studies that implicate COX-2 in cisplatin resistance. This context would help justify why COX-2 was selected for investigation.”*

Response: We appreciate the reviewer’s insightful comment. Cytochrome c oxidase (COX, complex IV) is the terminal complex of the mitochondrial respiratory chain, crucial for mitochondrial oxidative phosphorylation. COX subunits (e.g., COX6c, COX4) are implicated in cancer metabolism and drug resistance (PMID: 35879322, PMID: 25726526). Among 14 COX subunits, expression of the COX1 (MT-CO1) and COX2 (MT-CO2) subunits is known to be regulated by mitochondrial one-carbon metabolism and responsive to mitochondrial formyl donors, such as 10-formyl-THF (PMID: 29452640). Therefore, we assessed the levels of MT-CO1, MT-CO2, and COX4 and found that the specific subunit, MT-CO2, is altered by DLAT-MTHFD2 and is responsible for cisplatin resistance. This rationale and references are added to the revised manuscript (pages 10 and 28).

Note: We have used MT-CO2 instead of COX2 for clarity and distinction from cyclooxygenase 2, also known as COX2.

2. *“Page 7, Fig. 3: The role of mitochondrial ROS in chemotherapy resistance is complex. While the manuscript shows that DLAT influences ROS levels, it would benefit from more discussion on how ROS accumulation affects cellular processes beyond apoptosis.”*

Response: We thank the reviewer for raising this point. To investigate whether DLAT-regulated ROS influences cellular processes beyond apoptosis, we examined whether there are any additional changes in cellular events associated with ROS, including senescence, autophagy, and cell cycle progression, following DLAT loss and cisplatin treatment, and if they are restored when the elevated ROS is controlled by the antioxidant N-acetylcysteine (NAC). Unlike apoptosis shown in Fig. 3d, neither loss of DLAT nor NAC treatment dramatically affected the levels of p21Waf1/Cip1 (a senescence marker), LC3-I/II and p62 (autophagy markers), or cell cycle distribution (new Supplementary Fig. 3). These data suggest that DLAT-controlled ROS primarily supports cisplatin resistance through anti-apoptotic mechanisms.

New Supplementary Fig.3

3. “The data on DMp39 are promising (Fig. 8). However, the long-term stability and biodistribution of the peptide *in vivo* could be discussed in more depth. Furthermore, testing DMp39 in additional cancer types and combination regimens, would strengthen the therapeutic claims and highlight its potential clinical relevance.”

Response: We thank the reviewer for raising this point. To assess the *in vivo* biodistribution and stability of DMp39, we performed fluorescence imaging using cy7-labeled DMp39 in KB-3-1^{cisR}-bearing xenograft mice. Cy7-DMp39 was primarily localized to tumor tissues and remained detectable for up to 172 hours, suggesting prolonged tumor retention (Response Fig. 1a). *Ex vivo* imaging of major organs and tumors further confirmed that cy7-DMp39 largely localized to the tumors (Response Fig. 1b). Although some accumulation was observed in the liver, treatment with DMp39 and cisplatin induced minimal liver toxicity (new Fig. 8f left). These data support a favorable pharmacokinetic behavior of DMp39, which demonstrates selective accumulation and persistence in tumor tissues.

In addition, we tested the efficacy of DMp39 across multiple human cancer cell lines, including those of head and neck (FaDu), breast (MDA-MB231), cervical (HeLa), and ovarian (A2780) cancers. In all tested models, DMp39 in combination with cisplatin significantly enhanced apoptotic cell death and attenuated cell growth (new Supplementary Fig. 12a-d). These preclinical data suggest that DMp39 could be broadly applied as a chemotherapy sensitizer in various types of cancers.

Response Fig. 1

Response Fig. 1

b

New Figure 8f left

New Supplementary Fig.12

4. “Page 10-11, Fig. 7: The manuscript presents both pre- and post-treatment levels of DLAT-MTHFD2. It would be beneficial to clarify whether baseline levels or changes post-treatment are more critical for predicting cisplatin resistance. Additionally, the impact of cisplatin treatment on DLAT-MTHFD2 expression levels could also be explored.”

Response: We appreciate this insightful suggestion. We obtained additional tumor samples from HNSCC patients who received platinum-containing regimens. Together with the pre-existing stained tumors from the original manuscript, staining of additional cases allowed us to analyze 11 pre-treatment, 37 post-treatment, and 9 pre- and post-treatment pairs. Analysis of pre- and post-treatment paired tumor samples obtained from patients showed that the levels of DLAT, acetyl-MTHFD2, and MT-CO2 increased after cisplatin treatment (new Supplementary Fig. 10a-d). Moreover, post-treatment tumors from therapy-resistant patients showed significantly higher levels of DLAT signaling effectors than those from sensitive patients. In contrast, this difference was not observed in pre-treatment tumors (new Supplementary Fig. 10e-g).

Collectively, these results suggest that cisplatin treatment induces the levels of DLAT, acetyl MTHFD2, and MT-CO2, and these post-treatment levels, rather than the initial baseline levels, may be more critical for predicting cisplatin resistance.

New Supplementary Fig.10

These additional staining results were reflected when analyzing the link between expression and treatment outcomes (Revised Fig. 7a-b), and the correlation between DLAT and acetyl-MTHFD2 (Revised Fig. 7c) in the revised manuscript. Increased sample size further strengthened the significance of the finding.

Revised Figure 7a-c

5. “The study shows that DLAT is not involved in cisplatin resistance in colon cancer (Supplementary Fig. 2), despite its significant effects in lung cancer models. The authors could explore or provide more context on how the DLAT-MTHFD2 interaction is regulated across different cancer types. A deeper exploration into why DLAT is ineffective in colon cancer could enhance understanding of its specificity.”

Response: We thank the reviewer for bringing this important point to our attention. To further explore the cancer-type specificity of DLAT-MTHFD2 signaling, we analyzed the levels of DLAT and DLAT-mediated MTHFD2 acetylation in lung and colon cancer cells. DLAT expression in colon cancer was significantly lower compared to that in parental lung or cis^R cancer cell lines (Response Fig. 2a). More importantly, we found that although DLAT binds to MTHFD2 in all cell lines, MTHFD2 is not acetylated at K44 in the colon cancer cell lines tested. (new Supplementary Fig. 6c). Gene profile analysis using publicly available database indicated that DLAT gene level is significantly higher in lung adenocarcinoma compared to normal tissues, but this wasn’t the case for colon adenocarcinoma (Response Fig. 2b). We also performed a correlation analysis between the expression of 40 acetyltransferases and cisplatin resistance in lung cancer (n=97) or colon cancer (n=44) cell lines using DepMAP 24Q4 database (Response Fig. 2c). DLAT showed a weak yet positive correlation ($r = 0.16$) with cisplatin resistance (AUC) in lung cancer, but not in colon cancer ($r = -0.10$). Other acetyltransferases, such as NAGS ($r = 0.30$) or OGA ($r = 0.33$), showed greater correlation with cisplatin resistance in colon cancer cells.

Collectively, these data suggest that different acetylation-driven signaling among cancer types may influence cisplatin resistance. The role of DLAT in cancer may differ between lung and colon cancers, and DLAT-MTHFD2 may play a crucial role in cisplatin resistance in lung cancer. However, acetyltransferases such as NAGS or OGA, rather than the DLAT-MTHFD2 signaling pathway, may contribute to cisplatin resistance in colon cancer.

Response Fig. 2

New Supplementary Fig. 6c

6. “The manuscript suggests that DLAT is associated with post-target resistance to cisplatin. It would strengthen the paper if the authors could further explore and discuss post-target-related mechanisms, for example involving apoptosis-related proteins such as BCL-2 family members, or oxidative stress markers like Nrf2 and GSH.”

Response: We appreciate the reviewer's insightful suggestion. To further elucidate the downstream molecular mechanism by which the combination of cisplatin and DLAT knockdown enhances cell death, we assessed a panel of apoptotic factors and found that DLAT loss in cisplatin-treated cells specifically resulted in the reduction of anti-apoptotic protein Bcl-xL (new Fig. 3g). In contrast, control of elevated ROS with NAC in these cells partially restored Bcl-xL expression (new Fig. 3h). These data suggest that DLAT reduces ROS levels, thereby inducing Bcl-xL expression and contributing to the suppression of apoptotic cell death.

New Figures 3g and 3h

Inspired by the reviewer’s suggestion, we also assessed the effect of DLAT loss and cisplatin treatment on oxidative stress markers, including the GSH/GSSG ratio, NADPH level, and the expression and nuclear localization of Nrf2. In line with elevated ROS levels, impairment of DLAT resulted in a reduced/oxidized glutathione (GSH/GSSG) ratio and a lower NADPH level in cisplatin-treated cells (new Supplementary Fig. 2). DLAT loss or cisplatin treatment did not alter Nrf2 levels or nuclear localization in cisplatin-treated cells, suggesting that Nrf2 is not involved in DLAT-induced cisplatin resistance (Response Fig. 3). These data further suggest the interconnection between DLAT, MTHFD2, and redox homeostasis that involves NADPH, GSH/GSSG, and ROS.

New Supplementary Fig. 2

Response Fig. 3

7. “The mechanism by which DLAT selectively acetylates MTHFD2 over other mitochondrial proteins remains somewhat unclear. Providing additional analysis would further strengthen this key mechanistic insight.”

Response: Thank you for raising this point. We re-examined the DLAT interaction proteomics data, and we now present the top 10 mitochondrial candidates that most abundantly bind to DLAT beyond pyruvate dehydrogenase complex (PDC) components in the revised manuscript (new Fig. 4a). MTHFD2, PRDX3, MRPL family, and POLRMT were identified as potential interacting partners of DLAT (new Fig. 4c and Supplementary Fig. 4a). The interaction between DLAT and the four candidates in cancer cells was confirmed by co-immunoprecipitation (new Fig. 4b). However, the effect of gene target inhibition on cisplatin resistance revealed that loss of MTHFD2 or POLRMT (potential DLAT-MTHFD2 signaling effector as described below), but not the others, is involved in the cisplatin response (Fig. 4d and new Supplementary Fig. 4b-d). These studies further strengthen the rationale for selecting MTHFD2 over other mitochondrial proteins as a primary DLAT substrate that contributes to DLAT-induced cisplatin resistance.

We have newly identified that POLRMT, a critical component of the mitochondrial transcription initiation complex, is the downstream effector of DLAT and acetyl-MTHFD2, contributing to the induction of MT-CO2 (please see the new Supplementary Fig. 8 on page xv). It is plausible that the MTHFD2 product, 10-formyl-THF, and POLRMT coordinately promote the expression of MT-CO2, thereby conferring resistance to cisplatin.

New Figure 4a-b and Supplementary Fig. 4

4a

Protein identified in DLAT IP samples	A549 ^{cisR} +cisplatin	
	control	shDLAT
DLAT	97.8	1.0
MTHFD2	24.2	1.0
MRPL1	13.7	1.0
MRPL45	13.6	1.0
MRPL58	6.5	1.0
MRPL49	5.3	1.0
MRPL23	3.8	1.0
MRPL44	3.5	1.0
MRPL46	3.2	1.0
PRDX3	3.2	1.0
POLRMT	2.9	1.0

4b

4c

S4a

4d

S4b

S4c

S4d

REVIEWER 2

“The manuscript is well-written, with the strength of replicating experiments across multiple cell lines. The findings are novel and exciting, though some mechanistic gaps (outlined below) hinder full evaluation of their significance.”

1a. “In Figure 2F acetyl-CoA was added to cell media with the idea that this supplemented intracellular acetyl-CoA (measured by ELISA assay). This is not a valid experiment since acetyl-CoA is an intracellular metabolite and is not directly transported across the plasma membrane and acetyl-CoA measurements by ELISA assay are known to produce erroneous results (PMID: 37398224). Moreover, mitochondrial acetyl-CoA is separated from cytosolic acetyl-CoA (PMID: 36658431), so even if extracellular acetyl-CoA could indirectly supplement intracellular acetyl-CoA, there is no indication that mitochondrial acetyl-CoA is effected. The conclusion drawn from this experiment that effects of DLAT knockdown are unrelated to [acetyl-CoA] is not substantiated.”

Response: We thank the reviewer for raising this important point. As noted by the reviewer, the external supplementation of acetyl-CoA has limitations due to membrane impermeability and potential assay variability. To address this concern, we employed a series of alternative approaches involving the genetic modulation of mitochondrial enzymes involved in acetyl-CoA metabolism. We overexpressed acetyl-CoA producing enzyme, acetyl-CoA synthetase 1 (ACSS1), or knocked down acetyl-CoA consuming enzyme citrate synthase (CS) to load mitochondrial acetyl-CoA in the mitochondria (Response Fig. 4a). ¹³C-glucose tracing of these cells revealed that overexpression of ACSS1 but not CS knockdown accumulates mitochondrial acetyl-CoA in KB-3-1^{cisR} cells (Response Fig. 4b-c).

Based on this information, we decided to overexpress ACSS1 to provide mitochondrial acetyl-CoA in cells lacking DLAT. Instead of the ELISA assay, we performed LC-MS-based targeted metabolomics as newly described in the Methods section to detect intracellular acetyl-CoA levels. Although we successfully manipulated mitochondrial acetyl-CoA levels by ACSS1 overexpression, the total acetyl-CoA level did not change significantly, and it did not rescue the decreased cisplatin resistance caused by DLAT loss (new Fig. 2f). These data support the conclusion that the cisplatin sensitization observed upon DLAT knockdown is not attributable to any changes in the PDC product acetyl-CoA. Additional studies listed in response to comment 1b (pages x-xi) further demonstrate that the role of DLAT on MTHFD2 is PDC independent.

Response Fig. 4

New Figure 2f

1b. “The catalytic site deletion and mutation experiments (Fig 5F, Fig 8A) suggest that the lysine acetyltransferase mechanism is similar to acetyl transfer to CoASH. Is this activity sustained in the absence of the PDH and DLD subunits? The *in vitro* experiments (Fig 5A-E) suggest this subunit acts alone but PDH and DLD (and potentially other enzymes) may be associated and should be tested for. Direct comparison of WT and enzymatic mutated DLAT *in vitro* would clarify if the acetyl transferase activity can be directly attributable to DLAT rather than other (non PDC) associated enzymes.”

Response: We thank the reviewer for raising this point. To determine whether DLAT-mediated acetylation is dependent on other subunits of the pyruvate dehydrogenase complex (PDC), we first confirmed whether MTHFD2 interacts with PDC subunits, including DLAT, PDH, and DLD, in cisplatin-resistant cancer cell lines. Among these subunits, only DLAT, but not the other subunits, was found to associate with MTHFD2 (new Fig. 4g).

In addition, *in vitro* acetylation assays using purified flag-tagged DLAT variants and purified GST-fused MTHFD2 as a substrate demonstrated that DLAT wildtype (WT), but not its enzymatic mutants (S475A and ΔB), acetylates MTHFD2 in the absence of PDH and DLD (new Supplementary Fig. 5b), confirming that the acetyltransferase activity that led to MTHFD2 acetylation is mainly due to DLAT, rather than any other PDC or non-PDC enzymes associated with DLAT or MTHFD2. Furthermore, we assessed MTHFD2 acetylation in cancer cells under conditions where PDH or DLD were downregulated by knockdown. MTHFD2 acetylation level was increased by expression of DLAT WT but not by the enzymatic mutants, even in the absence of PDH or DLD subunit (new Supplementary Fig. 5c). This result further confirms that DLAT acetylates MTHFD2, and this non-classical acetyltransferase activity is independent of other PDC components in cisplatin-resistant cancer cells.

New Figure 4g

New Supplementary Fig. 5b and 5c

2. “Unanswered questions on the mechanism of action and specificity of the DMp39 peptide include: Is DMp39 acetylated as a competitive substrate for DLAT? Does DMp39 affect the acetyl-CoA generation activity of the PDC? Does the peptide affect 10-formyl-THF in vivo, as per the proposed mechanism? How does the peptide enter the mitochondria? It is surprising that toxicity markers do not respond to cisplatin (Figure 8E). How can these markers be appropriate measures of (lack of) toxicity of the peptide if they are unchanged by cisplatin treatment?”

Response:

i) To determine whether DMp39 functions as a competitive substrate, we incubated purified flag-DLAT and GST-fused MTHFD2 with increasing doses of DMp39. MTHFD2 K44 acetylation by DLAT was decreased by DMp39 in a dose-dependent manner, suggesting that DMp39 competes with MTHFD2 as a substrate for DLAT (new Fig. 8c).

New Figure 8c

ii) To determine whether DMp39 affects PDC activity, we assessed the phosphorylation status of PDHA1 at serine 293, a well-established inhibitory modification that correlates with PDC activity, as well as conducting a PDC activity assay. DMp39 treatment did not alter PDHA1 phosphorylation levels or PDC activity in A549^{cisR} and KB-3-1^{cisR} cells treated with cisplatin. This suggests that DMp39 does not influence PDC (new Supplementary Fig. 14a-b).

New Supplementary Fig. 14a-b

iii) To examine the effect of DMP39 on 10-formyl-THF levels in vitro and in vivo, we performed LC-MS-based targeted metabolomics to detect metabolite 10-formyl-THF in cells and *in vivo* in tumors collected from xenograft mice. In cell-based assays, 10-formyl-THF levels were significantly decreased upon DMP39 treatment (new Supplementary Fig. 14c). Consistently, 10-formyl-THF levels were significantly reduced in tumors collected from KB-3-1^{cisR} xenograft mice treated with DMP39 (new Supplementary Fig. 14d). These results together with other data shown in Fig. 8 demonstrate that DMP39 interferes in the DLAT-mediated MTHFD2 acetylation, which results in suppressed MTHFD2 activity shown as decreased MTHFD2 metabolic product 10-formyl-THF.

New Supplementary Fig. 14c-d

iv) How does DMP39 enter the mitochondria? We included the TAT sequence, YGRKKRRQRRR, after the MTHFD2 sequence, which is a cell-penetrating peptide that helps translocate DMP39 across cell membranes (Revised Fig. 8a). To test whether DMP39 localizes to the mitochondria, we performed line intensity profile analysis of Mitotracker (green) and cy7-DMP39 (red) signals to assess the colocalization. As shown by the overlap coefficients (*r*), we found that DMP39 is primarily localized in the mitochondria (new Supplementary Fig. 11a-b). DMP39 is derived from amino acids 39-66 of MTHFD2. This region exhibits physicochemical properties of an amphipathic α -helical structure. Notably, mitochondrial presequences often consist of amphipathic helices, which are known to mediate mitochondrial import through recognition by the translocase machinery. The TOM complex is a major part of this machinery. However, disruption of the TOM complex by Tom70 knockdown or treatment with the complex inhibitor t-2-hex (PMID: 40445107) did not interfere with DMP39 mitochondrial localization (Response Fig. 5a-b). This suggests that DMP39 can enter the mitochondria independently of the translocase machinery. Indeed, previous studies have shown that synthetic amphipathic peptides can enter mitochondria independently of the canonical translocase systems (PMID: 3396537). Thus, the amphipathic nature of the MTHFD2-derived sequence likely facilitates the mitochondrial localization of DMP39.

New Supplementary Fig. 11

Revised Fig.8a

a

b

Response Fig. 5

a

b

v) “It is surprising that toxicity markers do not respond to cisplatin (Figure 8E). How can these markers be appropriate measures of (lack of) toxicity of the peptide if they are unchanged by cisplatin treatment?”

In our *in vivo* study, we employed a sublethal dose of cisplatin (5 mg/kg, twice/week for 22 days) to model chronic exposure while minimizing systemic toxicity (PMID: 31081803). This dosing regimen is consistent with established protocols that aim to induce therapeutic effects without causing acute toxicity in mice (PMID: 34680523). To address concerns about the sensitivity of our toxicity markers, we conducted an additional experiment using a higher, acute dose of cisplatin (10 mg/kg, administered over 3 consecutive days) as a positive control. This higher dose is known to induce nephrotoxicity and systemic toxicity in mice. Following the administration of a lethal dose of cisplatin, we observed marked elevations in established toxicity markers, confirming their responsiveness under conditions of acute toxicity. Furthermore, we re-analyzed serum samples from our original experiments and confirmed that the toxicity markers remained at baseline levels in the sublethal cisplatin and/or DMp39 treatment groups, consistent with the absence of overt toxicity. In contrast, the acute high-dose cisplatin treatment induced a marked reduction in the spleen index and increased serum toxicity markers (new Fig. 8f). These findings validate the sensitivity and specificity of our toxicity markers in detecting cisplatin-induced toxicity.

New Figure 8f

3. *“There is no indication of how 10-formyl-THF is linked to COX2 induction. Metabolomic data would be useful to understand the specificity of the 10-formyl-THF response, and potential downstream affects. Figure 7 examines the association of DLAT and MTHFD2 with cisplatin resistance, additional analysis of COX2 and relationship to DLAT and MTHFD2 would be useful.”*

Response:

i) “How 10-formyl-THF is linked to MT-CO2 induction.” To gain mechanistic insight into how MT-CO2 (also known as COX2) is upregulated by 10-formyl-THF, we genetically downregulated four protein factors—POLRMT, TFAM, TFB2M, and NRF1—that are key regulators of mitochondrial gene expression and may induce MT-CO2 in cells. POLRMT but no other knockdowns reduced MT-CO2 levels in cells supplemented with 10-formyl-THF (new Supplementary Fig. 8a). Furthermore, treatment with POLRMT inhibitor IMT1 mitigated the MT-CO2 expression induced by 10-formyl-THF (new Supplementary Fig. 8b). POLRMT was identified as one of the top DLAT-interacting proteins (new Supplementary Fig. 4a on page viii). Its functional relevance was further confirmed, and its knockdown enhanced cisplatin-induced apoptosis and reduced cell viability (see new Supplementary Fig. 4d on page viii). These data collectively support that 10-formyl-THF promotes MT-CO2 expression in coordination with POLRMT, highlighting the mechanistic link between DLAT/MTHFD2-induced one-carbon metabolism and mitochondrial gene expression.

ii) *“Figure 7 examines the association of DLAT and MTHFD2 with cisplatin resistance. Additional analysis of COX2 and relationship to DLAT and MTHFD2 would be useful.”* We appreciate this insightful suggestion. As suggested, we stained MT-CO2 (a.k.a. COX2) in 48 head and neck squamous cell carcinoma (HNSCC) and 21 non-small cell lung carcinoma (NSCLC) patient tumors, which are the tumors we examined for DLAT and acetyl-MTHFD2. MT-CO2 staining results were similar to those of DLAT and acetyl-MTHFD2 in these tumors, showing MT-CO2 levels higher in the resistant groups than in the sensitive groups of HNSCC and NSCLC patients who received chemotherapy-containing regimens (Supplementary Fig. 9a-b).

We also analyzed paired pre- and post-treatment tumor samples and found that the levels of DLAT signaling effectors, including MT-CO2, increased after cisplatin treatment (new Supplementary Fig. 10a-d). Moreover, post-treatment tumors from therapy-resistant patients showed significantly higher levels of MT-CO2 than those from sensitive patients. In contrast, this difference was not observed in pre-treatment tumors (new Supplementary Fig. 10g). These data suggest that cisplatin treatment induces the levels of DLAT signaling effectors, including MT-CO2, and these post-treatment levels, rather than the initial baseline levels, may be more critical for predicting cisplatin resistance.

Lastly, we assessed the relationship of MT-CO2 to DLAT and acetyl-MTHFD2 in these tumors. We observed a relatively strong positive correlation between MT-CO2 and DLAT or acetyl-MTHFD2 levels in tumor specimens collected from patients with HNSCC who received platinum-based chemotherapy (MT-CO2 vs. DLAT: Pearson correlation, $r = 0.536$; MT-CO2 vs. acetyl-MTHFD2: $r = 0.503$) (new Supplementary Fig. 9c). Tumors from NSCLC patients who received chemioimmunotherapy showed weaker but still positive correlations (MT-CO2 vs. DLAT: $r = 0.177$; MT-CO2 vs. acetyl-MTHFD2: $r = 0.298$) (new Supplementary Fig. 9d). It is plausible that additional factors beyond DLAT-MTHFD2 may contribute to the regulation of MT-CO2 expression in NSCLC. Collectively, these additional studies support the clinical relevance of DLAT-acetyl-MTHFD2-MT-CO2 in cisplatin resistance.

New Supplementary Fig.9a-b

New Supplementary Fig.10a, 10d, 10g

New Supplementary Fig.9c and 9d

Minor comments:

4. “Figure 2E: evidence that DCA is effective should be provided i.e. phospho PDH blots, metabolic tracing.”

Response: Thank you for pointing this out. We included phospho-PDHA1 S293 blots to show that DCA treatment was effective in A549^{cisR} and KB-3-1^{cisR} cells (Revised Fig. 2e).

Revised Figure 2e

5. “Figure 4F: MTHFD2 IP: blotting of other PDC subunits is required to understand the specificity of the interaction with DLAT. A negative control would help to evaluate the IP.”

Response: As suggested, we newly performed MTHFD2 co-immunoprecipitation in A549^{cisR} and KB-3-1^{cisR} cells to detect not only DLAT but also other PDC subunits, PDH, and DLD. Among PDC subunits, only DLAT, but not the other subunits, specifically bound to MTHFD2. We also included a negative control, rabbit IgG (new Fig. 4g - old Fig.4f).

New Figure 4g (old Figure 4f)

6. “Schematic illustrations would be beneficial to illustrate PDC complex with each subunit labelled (Figure 2) and acetylation and mutation and sites (Figure 5).”

Response: As suggested, we included a schematic illustration of the PDC complex with subunits labeled and acetylation and mutation sites of MTHFD2 in the revised Fig. 2d and Fig. 5b, respectively.

Revised Figure 2d

Revised Figure 5b

7. “Some panels are too small to read e.g Fig 4A, 5C chromatograms and Fig. 8A model.”

Response: As suggested, we modified the figure layout and size for Figures 4a, 4c, 5c, and 8a to enhance visual clarity.

Old Figure 4a

Revised Figure 4a

Protein identified in DLAT IP samples	A549 ^{cisR} +cisplatin	
	control	shDLAT
DLAT	97.8	1.0
MTHFD2	24.2	1.0
MRPL1	13.7	1.0
MRPL45	13.6	1.0
MRPL58	6.5	1.0
MRPL49	5.3	1.0
MRPL23	3.8	1.0
MRPL44	3.5	1.0
MRPL46	3.2	1.0
PRDX3	3.2	1.0
POLRMT	2.9	1.0

Old Figure 5c

Revised Figure 4c and new Supplementary Fig.4a

Old Figure 8a

Revised Figure 8a and Supplementary Fig.5a

8a

S5a

REVIEWER 3

“The manuscript is well-written with clear logic and seamless flow. Experimental methods were provided in detail and the data were clearly presented with logical interpretations. The overall conclusions of the manuscript are supported by the extensive data in the results section. The work presented in the manuscript is of significance to the field of anticancer research as the newly identified mechanism of drug resistance has important therapeutic implications.”

“The only suggestion is to assess the synergistic effects of cisplatin and a COX2 inhibitor (either in vitro or in vivo) given that very selective and potent COX2 inhibitors are available.”

Response: We appreciate the reviewer's insightful suggestion and sincerely apologize for not making the term clearer. COX2 in the manuscript refers to ‘mitochondrially encoded cytochrome C oxidase II’, not ‘cyclooxygenase-2’. We have changed 'COX2' to 'MT-CO2' in the revised manuscript for clarity and to distinguish it from cyclooxygenase 2, also known as COX2.

While several potent cyclooxygenase-2 inhibitors, such as celecoxib, valdecoxib, and etoricoxib, are available, there are currently no selective pharmacological inhibitors targeting MT-CO2. Thus, we were unable to assess the effects of a COX2 inhibitor on cisplatin response. However, we were able to obtain siRNAs for cytochrome C oxidase II to test whether targeting this DLAT-acetyl-MTHFD2 downstream effector sensitizes cisplatin-resistant cancer cells to cisplatin. As shown in the new Supplementary Figure 7, MT-CO2 was successfully downregulated using two independent siRNAs (new Supplementary Fig. 7a). It sensitized cisplatin-resistant cancer cells to cisplatin, and thereby inducing enhanced apoptotic cell death and decreased cell viability (new Supplementary Fig. 7b). We also stained MT-CO2 using primary tumors collected from head and neck cancer or lung cancer patients and demonstrated the association of MT-CO2 with cisplatin resistance and correlation between MT-CO2 and its upstream effectors, DLAT or acetyl-MTHFD2 (new Supplementary Fig. 9a-d, 10a right, 10d, and 10g; please see pages xvi-xvii). Collectively, these data suggest that DLAT-acetyl-MTHFD2-MT-CO2 signaling confers cisplatin resistance, and this signaling axis could be targeted using the mechanism-based drug Dmp39.

New Supplementary Fig. 7

Response to Review

We sincerely appreciate your thoughtful review of our revised manuscript (NCOMMS-24-55974A). We have addressed these insightful and constructive comments from the Reviewers as detailed below, and we believe that the manuscript has been further strengthened as a consequence of this revision.

Reviewer #1

1. The newly included in vivo data on tumor-selective accumulation and stability of Cy7-DMp39 enhance the translational relevance of the study. Additionally, the use of multiple cancer models to validate DMp39 efficacy provides encouraging breadth. It may be valuable to briefly discuss the potential for future testing in resistant, patient-derived models across a wider range of tumor lineages.

Response: We thank the reviewer for highlighting the translational significance of our *in vivo* findings. In accordance with the reviewer's suggestion, we have expanded the Discussion section as listed below (page 17).

“To enhance translational relevance, future studies should evaluate DMp39 in patient-derived xenograft and organoid models that better capture clinical heterogeneity, as well as across a broader range of tumor types.”

2. While the decoy peptide DMp39 demonstrates promising activity in preclinical models, a more detailed discussion of its pharmacokinetics and pharmacodynamics (e.g., half-life, clearance, bioavailability) would help contextualize its clinical potential and inform prospective dosing strategies. Similarly, although minimal toxicity has been observed, data from longer-term toxicity studies (e.g., over 4–6 weeks) would further support the safety profile, especially in light of the peptide's mitochondrial localization.

Response: We appreciate the reviewer's insightful suggestions regarding the clinical potential of DMp39. In response, we have incorporated a discussion on the pharmacokinetic and pharmacodynamic considerations of TAT-fused peptides, including anticipated rapid systemic clearance, short plasma half-life, and limited bioavailability. We also discussed strategies such as PEGylation or backbone cyclization to enhance peptide stability and reduce immunogenicity. Furthermore, we acknowledged the need for extended safety studies to fully evaluate the long-term toxicity of DMp39 given its mitochondrial localization. These points are now included in the Discussion section as listed below (pages 17 and 18).

“While DMp39 exhibited robust anti-tumor efficacy in multiple preclinical models with minimal acute toxicity, further pharmacokinetic and pharmacodynamic evaluations are necessary to advance clinical translation. As with other TAT-fused or cationic peptides, rapid systemic clearance and short plasma half-life are anticipated, which may limit bioavailability and therapeutic durability. Structural modifications such as lipidation, PEGylation, or backbone cyclization may enhance the peptide's *in vivo* stability and reduce proteolytic degradation. These strategies may also mitigate potential immunogenicity commonly associated with TAT-fused or cationic peptides, thereby addressing key translational barriers to long-term therapeutic use. Given the mitochondrial localization of DMp39, it is also critical to assess long-term safety. Although our short-term dosing of DMp39 showed no acute toxicity, extended studies will be required to rule out delayed or cumulative toxicity in vital organs. These efforts will inform rational dosing strategies and ensure safety in future translational applications. Collectively, our finding provides a compelling rationale for applying DMp39 as a targeted mitochondrial therapeutic to overcome chemotherapy resistance, and further optimization is warranted to advance DMp39 toward clinical development.”

3. *Figure 8g would be strengthened by the inclusion of quantitative analysis to accompany the representative images.*

Response: We thank the reviewer for this helpful suggestion. In response, we have provided a quantitative analysis in Figure 8g.

4. *Given the reliance on TAT-based delivery for DMp39, it would be helpful to include a brief comment on potential immunogenicity or other translational barriers associated with peptide-based therapeutics.*

Response: We thank the reviewer for this important point. In response, we have added a discussion regarding the potential immunogenicity and other translational barriers associated with TAT-fused or cationic peptides in the Discussion section. We also described strategies such as PEGylation or structural modification that may reduce immunogenicity and enhance the translational potential of DMp39, as shown in response to comment #2 above (pages 17 and 18).

Reviewer #2

1. *The revised manuscript is significantly improved, and my previous concerns were thoroughly addressed. In particular, the in vitro acetylation assay (Supp. Fig. 5b) clearly demonstrates that wild-type (WT), but not mutant, DLAT efficiently catalyzes the acetylation of MTHFD2 at K44. I suggest that Supp. Fig. 5b replace the current main Fig. 5a, as it represents a stronger experiment with appropriate controls and effectively shows that acetylation is not dependent on the presence of acetyl-CoA.*

Response: We thank the reviewer for this valuable suggestion. In accordance with the recommendation, we have moved the original Figure 5a to Supplementary Figure 5a and relocated the previous Supplementary Figure 5b to the main figure panel as the new Figure 5e. This revised figure clearly demonstrates that MTHFD2 acetylation by DLAT occurs independently of acetyl-CoA. All relevant figure numbering and legends have been updated accordingly.

2. *More detail in methods are required for in vitro acetylation assay, MTHFD2 activity assay, and proteomics assay which are missing details on cell lysis and immunoprecipitation conditions or appropriate references to such details, and for PDC activity assay.*

Response: We appreciate the reviewer's attention to methodological clarity. In response, we have revised the Methods section to include additional details such as specific conditions for the in vitro acetylation assay, MTHFD2 activity assay, proteomics assay, PDC activity assay, and quantification of acetyl-CoA and 10-formyl-THF by LC-MS (pages 22-25).

3. *Figure 3h stroke missing from +*

Response: We appreciate the reviewer's attention to detail. Figure 3h (new Figure 3i) has been corrected, and all data points now have an appropriate stroke, as they did in the original manuscript.

4. *Figure 4a samples typo*

Response: We thank the reviewer for pointing this out. The typographical error in Figure 4a has been corrected.